

# On the morphological variability of *Ichniotherium* tracks and evolution of locomotion in the sistergroup of amniotes

Michael Buchwitz[1,*] and Sebastian Voigt[2,*]

[1] Museum für Naturkunde Magdeburg, Magdeburg, Germany
[2] Urweltmuseum Geoskop, Kusel, Germany
* These authors contributed equally to this work.

Corresponding authors
Michael Buchwitz,
michael.buchwitz@
museen.magdeburg.de
Sebastian Voigt,
s.voigt@pfalzmuseum.bv-pfalz.de

## ABSTRACT

*Ichniotherium* tracks with a relatively short pedal digit V (digit length ratio V/IV < 0.6) form the majority of yet described Late Carboniferous to Early Permian diadectomorph tracks and can be related to a certain diadectid clade with corresponding phalangeal reduction that includes *Diadectes* and its close relatives. Here we document the variation of digit proportions and trackway parameters in 25 trackways (69 step cycles) from nine localities and seven further specimens with incomplete step cycles from the type locality of *Ichniotherium cottae* (Gottlob quarry) in order to find out whether this type of *Ichniotherium* tracks represents a homogeneous group or an assemblage of distinct morphotypes and includes variability indicative for evolutionary change in trackmaker locomotion. According to our results, the largest sample of tracks from three Lower Permian sites of the Thuringian Forest, commonly referred to *I. cottae*, is not homogeneous but shows a clear distinction in pace length, pace angulation, apparent trunk length and toe proportions between tracks from Bromacker quarry and those from the stratigraphically older sites Birkheide and Gottlob quarry. Three Late Carboniferous trackways of *Ichniotherium* with relatively short pedal digit V from Haine's Farm, Ohio, and Alveley near Birmingham, United Kingdom, that have been referred to the ichnotaxa "*Baropus hainesi*," "*Megabaropus hainesi*" and "*Ichniotherium willsi*," respectively, share a marked outward rotation of foot imprints with respect to walking direction. Apart from this feature they are in many aspects similar to the Birkheide and Gottlob records of *I. cottae*. With the possible exception of the Maroon Formation (Early Permian, Colorado) sample, Early Permian *Ichniotherium* trackways with a relatively short pedal digit V fall into the morphological spectrum of the three well defined "Hainesi–Willsi," "Birkheide–Gottlob" and "Bromacker" morphotypes. With their more obtuse pace angulations and higher body-size-normalized pace and stride lengths the Bromacker type tracks imply higher walking speeds of their trackmakers compared to all other *Ichniotherium* tracks. More generally, a trend towards higher locomotion capability from the last common ancestor of all *Ichniotherium* producers to the last common ancestor of all "*Ichniotherium* with relatively short pedal digit V" and from the latter to the trackmakers of the mid-Early Permian Bromacker type can be deduced—with the reservation that overall sample size is relatively small, making this scenario a preliminary assessment. Whether the presumed advancements represent a more general pattern within diadectomorphs remains open until the non-European *Ichniotherium* trackway record improves. Ichnotaxonomic implications are discussed.

Pace angulation, Step cycle, Ichnology, Multivariate statistics

## INTRODUCTION

*Ichniotherium Pohlig, 1892* is a common and widespread kind of Late Carboniferous and
Early Permian tetrapod footprints referred to diadectomorph trackmakers (*Haubold,
2000*; *Voigt & Haubold, 2000*; *Voigt, 2005*, *2012*; *Voigt, Small & Sanders, 2005*; *Voigt,
Berman & Henrici, 2007*; *Voigt et al., 2011*, *2012*; *Voigt & Ganzelewski, 2010*; *Brink,
Hawthorn & Evans, 2012*; *Voigt & Lucas, 2015*, *2017, in press*). Tracks of this type were
discovered for the first time in Early Permian continental deposits of the Thuringian
Forest, central Germany (*Cotta, 1848*). During the last 150 years, *Ichniotherium*
tracks have been given at least 10 different ichnogeneric, 10 ichnospecific and 11
ichnosubspecific names (see *Voigt, 2005*, Appendix 15). A rather recent careful revision of
the central European *Ichniotherium* record (*Voigt, 2005*; *Voigt, Berman & Henrici, 2007*;
*Voigt & Ganzelewski, 2010*) argued for a single ichnogenus and three valid ichnospecies:
*Ichniotherium cottae* (*Pohlig, 1885*), *Ichniotherium sphaerodactylum* (*Pabst, 1895*) and
*Ichniotherium praesidentis* (*Schmidt, 1956*).

   *Ichniotherium cottae* on the one hand and *I. sphaerodactylum* and *I. praesidentis* on
the other are separated by pedal digit proportions considering that pedal digit V is
about as long as pedal digit II in *I. cottae* whereas it is usually as long as pedal digit III in
the two remaining ichnospecies (*Voigt & Ganzelewski, 2010*). A quantitative expression of
this distinction criterion is the pedal digit length ratio IV/V and a linear discriminant function
based on these digit lengths ($F = 1.2264 \times p_{IV} - 1.9728 \times p_V - 3.48281$; Fig. 1; Supplemental
Information S1). *I. sphaerodactylum* and *I. praesidentis*, which differ considerably in the
imprint morphology of the manus as well as in the trackway pattern, are very rare and
have only been recorded from central Germany (*Voigt, 2005*; *Voigt & Ganzelewski, 2010*),
Morocco (*Voigt et al., 2011*) and Canada (*Brink, Hawthorn & Evans, 2012*).

   *Ichniotherium* tracks with relatively short pedal digit V (length ratio $p_V/p_{IV} < 0.6$)
are much more common with undoubted occurrences from the Czech Republic
(*Fritsch, 1887*, *1895*, *1912*), Germany (*Pabst, 1908*; *Haubold & Stapf, 1998*; *Voigt &
Haubold, 2000*; *Voigt, 2005*, *2012*), Great Britain (*Haubold & Sarjeant, 1973*, *1974*),
Morocco (*Lagnaoui et al., in press*), Poland (*Pabst, 1908*; *Voigt et al., 2012*) and the United
States of America (*Carman, 1927*; *Baird, 1952*; *Voigt, Small & Sanders, 2005*; *Voigt &
Lucas, 2015, in press*). During the last decade numerous additional finds and yet
unpublished revision studies of previously recorded but misidentified specimens
significantly extended the global record of *Ichniotherium* tracks with relatively short
pedal digit V. Among these records only some include a notable sample of actual
trackways, i.e., imprint sequences comprising one or several step cycles—a necessary
precondition for their consideration in this approach which follows the directive that
taxonomy of vertebrate tracks shall not merely be based on imprint morphology but
take into account trackway features.

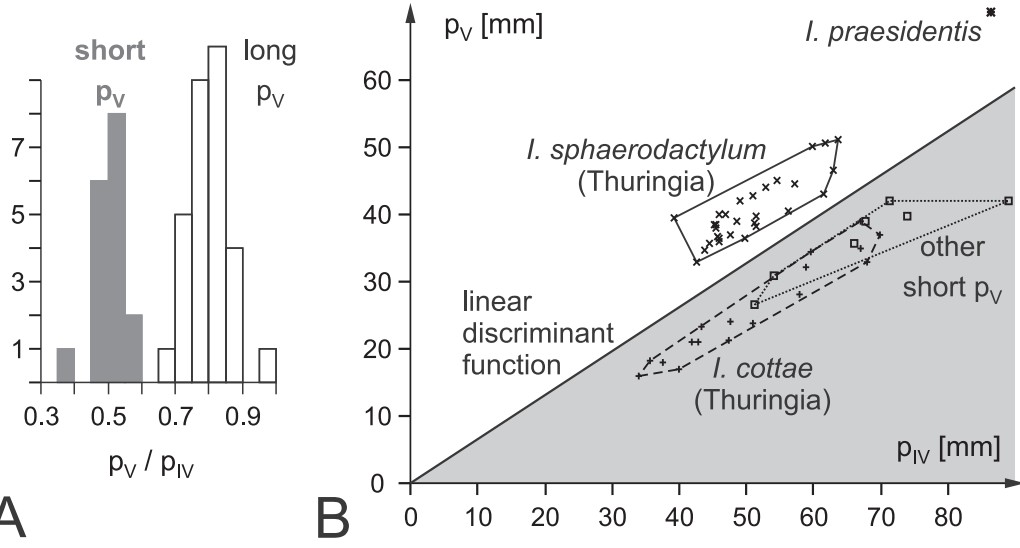

**Figure 1 Ichnotaxonomic composition of *Ichniotherium* and the variability in pedal toe lengths IV and V as a diagnostic criterion.** (A) Histogram depicting the distribution of pedal digit ratios V/IV in *Ichniotherium cottae* from the Thuringian Forest (gray) and the other German records, *I. sphaerodactylum* and *I. praesidentis* (white). (B) Distribution of *I. sphaerodactylum, I. praesidentis, I. cottae* and further records of *Ichniotherium* in a plot of pedal digit length V against pedal digit length IV (see Supplemental Information S1).

With some exceptions (*Ruta & Coates, 2007*), phylogenetic analyses of basal amniotes and their non-amniote relatives recovered the diadectomorphs as a monophylum that forms the sistergroup to amniotes and consists of *Limnoscelis, Tseajaia* as well as five or more diadectid taxa that range from the Late Carboniferous through the late Early Permian (*Laurin & Reisz, 1999*; *Ruta, Coates & Quicke, 2003*; *Reisz, 2007*; *Kissel, 2010*; Fig. 2). If the assignment of a fragmentary skull from China to diadectids by *Liu & Bever (2015)* is confirmed, it would extend the range of this group by over 15 million years into the Late Permian. Based on the unique track-trackmaker co-occurrences of the Early Permian Bromacker site in central Germany (*Voigt, Berman & Henrici, 2007*), *I. sphaerodactylum* with relatively long pedal digit V can be related to *Orobates pabsti* (*Berman et al., 2004*), whereas *I. cottae* with relatively short pedal digit V is likely to be the track of *Diadectes absitus* (*Berman, Sumida & Martens, 1998*), which has been referred to a new genus by *Kissel (2010)*. A short pedal digit V and phalangeal formula 2-3-4-5-3 has been documented for the North American *Diadectes* specimen CM 41700 (and was probably also present in *Diadectes absitus*, see *Voigt, Berman & Henrici, 2007*) whereas the relatively long pedal digit V and pedal phalangeal formula 2-3-4-5-4 occurs in both, the basal diadectomorph *Limnoscelis* (*Kennedy, 2010*, Fig. 8) and the diadectid *Orobates*. Thus, we consider a clade of diadectids which share a pes with relatively short digit V and are more closely related to the North American species of *Diadectes* (sensu *Kissel, 2010*) than to *Orobates* as the potential producer group of all documented *Ichniotherium* tracks with relatively short pedal digit V. This type of *Ichniotherium* represents a well-defined subset of all *Ichniotherium* tracks

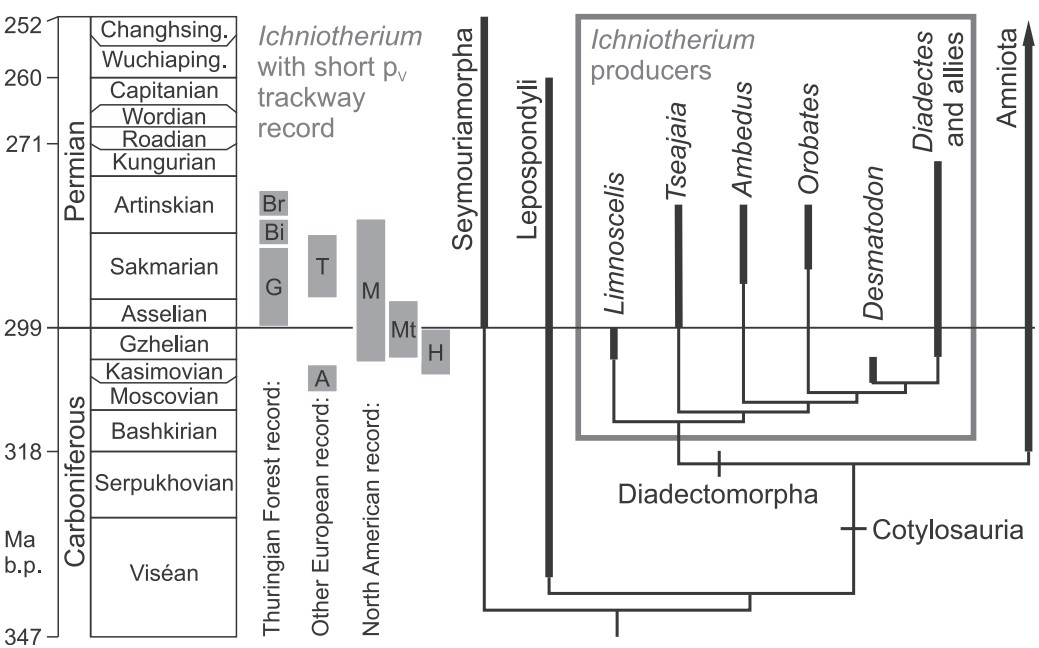

**Figure 2 Fossil record of *Ichniotherium* trackways with relatively short pedal digit V and phylogeny of the Diadectomorpha within derived reptiliomorphs.** Localitiy abbreviations: A, Alveley; Bi, Birkheide; Br, Bromacker; G, Gottlob type locality; H, Haine's Farm; M, Maroon; Mt, Marietta; T, Tłumaczow. Modified after *Reisz (2007)*, *Kissel (2010)* and *Voigt & Ganzelewski (2010)*.

(including *I. cottae* from the Thuringian Forest) and spans 20 million years of the geological record.

In order to find out whether *Ichniotherium* with relatively short pedal digit V, (a) can be subdivided into ichnotaxonomically relevant morphotypes based on imprint and trackway measurement data and (b) includes variability indicative for evolutionary change in trackmaker locomotion, the following steps are undertaken in the present approach:

(1) Documentation of the variability of imprint and trackway measures in *I. cottae* tracks from three localities of the Thuringian Forest in central Germany, i.e., Gottlob/ Friedrichroda (*I. cottae* type locality), Birkheide/Tambach-Dietharz and Bromacker/ Tambach-Dietharz as a reference sample for *Ichniotherium* with relatively short pedal digit V.

(2) Comparison of the Thuringian Forest record to Late Carboniferous and Permian records from Europe and North America, and, if feasible, distinction of trackway morphotypes. Taking the varying sample sizes into account, numerical discrimination schemes shall be derived for the largest samples and are then to be used for the classification of isolated trackways.

(3) Discussion of ichnotaxonomic consequences for *Ichniotherium* with relatively short pedal digit V.

(4) Inference of evolutionary change in trackway and imprint measures based on a phylogenetic hypothesis of *Ichniotherium* trackmaker relationships and discussion of individual track-trackmaker assignments.

(5) Inference of evolutionary change in diadectomorph locomotion based on phylogenetic trends in functionally relevant trackway/imprint measures and discussion of factors limiting interpretation at the present state of knowledge.

## MATERIALS AND METHODS

### Material

Based on the distinction criterion given above (pedal digit length ratio <0.6; $F > 0$, see Supplemental Information S1) only trackways with at least one step cycle preserved are considered herein. Because of the sparse trackway record from the *I. cottae* type locality Gottlob quarry (Thuringian Forest, central Germany) we have made an exception and included seven specimens with incomplete *Ichniotherium* step cycles (HF 57, HF 89, MNG 1381, MNG 1382, MNG 1385, MNG 1387, MNG 1781; Fig. 3) from this site. The present analysis includes the following 25 *Ichniotherium* specimens that include at least one complete step cycle (Table 1; Supplemental Information S2; Figs. 4–10): BU 2471, CMNH VP-3052, DMNS 50618, DMNS 50622, DMNS 55056, KGM-1, MB.ICV.3-F1, MB. ICV.3-F2, MC-1, MNG 1352, MNG 1386-F1, MNG 1819, MNG 2047, MNG 2049, MNG 2356-16-F1, MNG 2356-16-F2, MNG 10179, MSEO-I-36, NHMS AP-244-19, NHMS P-418, OSU 16553, PMJ P-1321-F3, SSB-1, UGKU 130-F1, UGKU 130-F2. All materials have been studied, documented und measured by one of us (SV) between 1998 and 2015.

### Use of imprint and trackway parameters

Considering their robustness as imprint measures, we focus here on digit proportions as the sole criterion for imprint shape. Length of pedal digit IV, usually the longest toe of an imprint pair, is used as a proxy for body size and for normalization of other toe lengths: $p_I(n) = p_I/p_{IV}$, $p_{II}(n) = p_{II}/p_{IV}$, ..., $m_V(n) = m_V/p_{IV}$. If only manual imprint proportions are considered we also use the ratios $m_I/m_{IV}$, $m_{II}/m_{IV}$, $m_{III}/m_{IV}$ and $m_V/m_{IV}$ (Supplemental Information S3; Fig. 11A). Concerning the use of pedal and manual digit length IV as normalization values we are following the convention of earlier studies (*Voigt, Berman & Henrici, 2007*; *Voigt & Ganzelewski, 2010*; contra *Romano & Citton, 2015*). Both measures are highly correlated with the average of all other toe lengths (correlation coefficients between 0.968 and 0.993) pedal digit length IV also shows high correlation with pes length (correlation coefficients between 0.930 and 0.964; see Fig. S1) and thus can be considered as a reasonable body size proxy.

Despite the uncertainties that can occur in free digit length measurements (Figs. 11B–11D), pedal and manual digit length IV are usually better recognizable than the overall imprint length which is often obscured by an indistinctly preserved posterior margin.

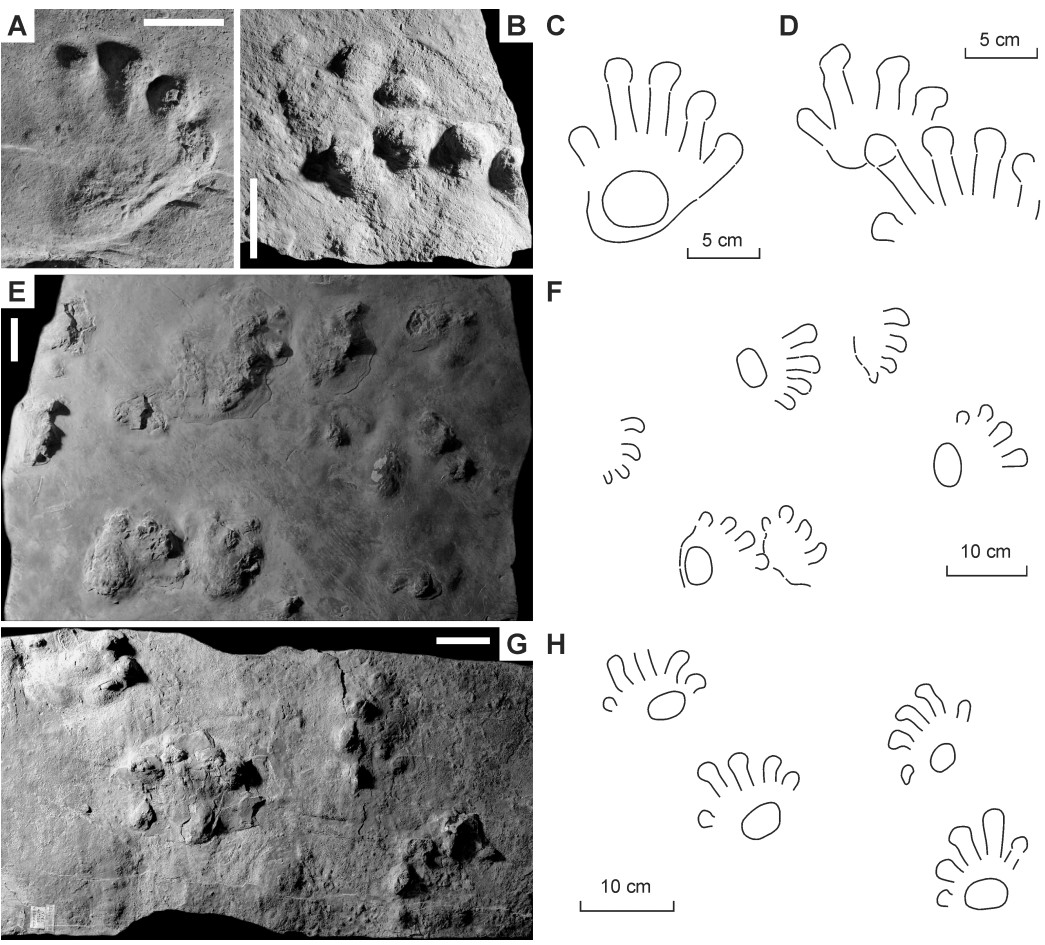

**Figure 3 _Ichniotherium cottae_ from the Early Permian Goldlauter Formation (Gottlob locality, Thuringian Forest, Germany).** (A–C) HF 57; (B, D) HF 89; (E, F) MNG-1386; (G, H) MNG-1781. Unlabeled scale bars equal 5 cm.

In order to compare trackway patterns in detail we consider each step cycle of a trackway as an individual dataset whose important attributes include the manual and pedal pace angulations ($\alpha_p$ and $\alpha_m$), the deviation of manual and pedal imprint orientations from walking direction ($\beta_p$ and $\beta_m$, positive value = inward rotation = dextral rotation of left hand/foot = sinistral rotation of the right hand/foot), normalized pace length ($P_p(n)$ = average of two pedal pace lengths/length of pedal digit IV), normalized apparent trunk length ($C(n)$ = apparent trunk length/$p_{IV}$; see scheme in Fig. 11A; Supplemental Information S4). Although they are redundant measures that can be calculated from normalized pace length and pace angulation in case of a regular trackway pattern, we also take the normalized stride length and trackway width ($S_p(n)$ = pedal stride length/$p_{IV}$; $B_p(n)$ = pedal trackway gauge width/$p_{IV}$) into account since they represent useful indicators for functional variation in track producers. For the normalization of trackway measures $P_p(n)$, $C(n)$, $S_p(n)$ and $B_p(n)$ the trackway average of pedal digit length IV is used.

**Table 1  Structure of the trackway dataset.**

| Locality | Stratigraphy | Number | Previous assignment (author) |
|---|---|---|---|
| Gottlob/Friedrichroda, Thuringia, Germany | E. Permian (Asselian–Sakmarian) | 8/2/12 | *I. cottae* (*Voigt, 2005*) |
| Birkheide/Tambach, Thuringia, Germany | E. Permian (Sakmarian–Artinskian) | 4/11/19 | *I. cottae* (*Voigt, 2005*) |
| Bromacker/Tambach, Thuringia, Germany | E. Permian (Artinskian) | 12/32/56 | *I. cottae* (*Voigt & Haubold, 2000*; *Voigt, 2005*; *Voigt, Berman & Henrici, 2007*) |
| Haine's Farm, Ohio, USA | L. Carboniferous (Kasimovian–Gzhelian) | 2/9/13 | *Megabaropus hainesi* (*Baird, 1952*) |
| Alveley/Shropshire, UK | L. Carboniferous (Moscovian–Kasimovian) | 1/4/6 | *I. willsi* (*Haubold & Sarjeant, 1973*) |
| Maroon Formation, Colorado, USA | L. Carboniferous (Moscovian)– E. Permian (Asselian) | 3/6/15 | *I. cottae* 2005 (*Voigt, Small & Sanders, 2005*) |
| Marietta, Ohio, USA | L. Carboniferous (Gzhelian)— E. Permian (Asselian) | 1/3/5 | *Ichniotherium* cf. *cottae* (unpublished) |
| Tłumaczow, Poland | E. Permian (Sakmarian) | 1/2/4 | *I. cottae* (*Voigt et al., 2012*) |
| **Total** | **Late Carboniferous (Moscovian)– Early Permian (Artinskian)** | **32/69/130** | **Three ichnospecies** |

**Note:**
Numbers refer to specimens, step cycles and imprint pairs (or individual imprints if only one of a pair is preserved; see also Supplemental Information S2).

## Quantitative comparison of trackway records

Toe proportions and trackway parameters are analyzed as separate datasets. All localities are represented by sets of imprint pairs and step cycles, whose toe proportions and trackway parameters are compared—mainly through methods of multivariate statistics for which the statistical software package PAST is used (*Hammer, Harper & Ryan, 2001*). We use bivariate plots and principal component analysis (PCA) to explore the datasets for noteworthy differences between sampled localities and multivariate analysis of variance (MANOVA) to test whether the supposed differences between localities are statistically significant. It should be noted that the PCA results depicted in this manuscript are based on (a) covariance matrices of length ratios (normalized length measurements) or (b) correlation matrices of length ratios in combination with angle measurements. The representation of morphospace in our PC plots differs from that of PC plots based on landmark analysis that are used in geometric morphometrics. In addition to toe-ratio-based analyses we have also carried out PCAs based on logarithmized toe ratios which might reveal body-size related biasing in the distribution and overlap of groups (Fig. S2). If the separability between groups appears to be good enough we employ linear discriminant analysis—preferably based on a small set of parameters—to gain a linear discriminant function for classification of further individual tracks and smaller track records, e.g., the trackways from Tłumaczow/Poland and Marietta/Ohio (according to the previously found morphotypes).

Usually not all toes of an imprint pair are preserved well enough to be measured, often the lateral digits (pedal and manual digit V) are missing or their connection with the sole imprint is vague. Thus, for reasons of sample size, hand proportions ($m_I(n)$ to $m_V(n)$) and toe proportions ($p_I(n)$ to $p_V(n)$) are compared separately and only the proportions of the more often preserved manual and pedal digits are combined in multivariate analyses.

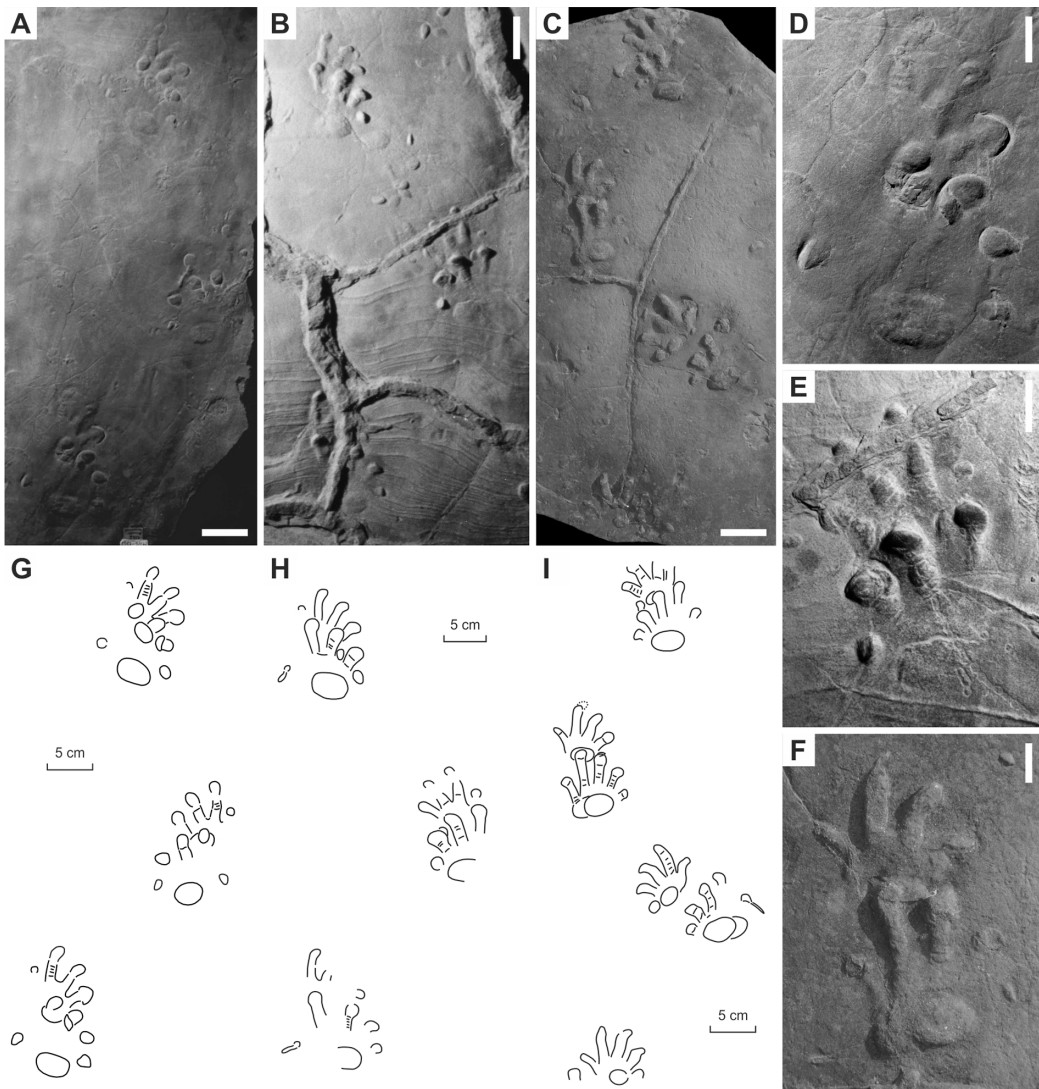

**Figure 4** *Ichniotherium cottae* from the Early Permian Tambach Formation (Bromacker locality, Thuringian Forest, Germany). (A, D, G) MNG 1352; (B, E, H) MB.ICV.3-1; (C, F, I) SSB-1. Unlabeled scale bars equal 5 cm (A–C) and 2 cm (D–F).               

## RESULTS

### Variation within the sample of *I. cottae* from the Thuringian Forest

Two separate principal component analyses that include (a) all pedal toe proportions (Fig. 12A) and (b) all manual toe proportions (Fig. 12B) show markedly deviant distributions for the Bromacker and Gottlob samples and for the Bromacker and Birkheide samples, respectively, within the first two principal components (plane of greatest variance); the distribution for the third locality lies in between (similar results for logarithmized toe ratios, see Fig. S2). According to average toe proportions (Table 2) the Bromacker tracks feature imprints with relatively short marginal digits $p_I$, $p_V$ and $m_I$ and relatively large manual imprints (higher ratio of manual digit IV to pedal digit IV)

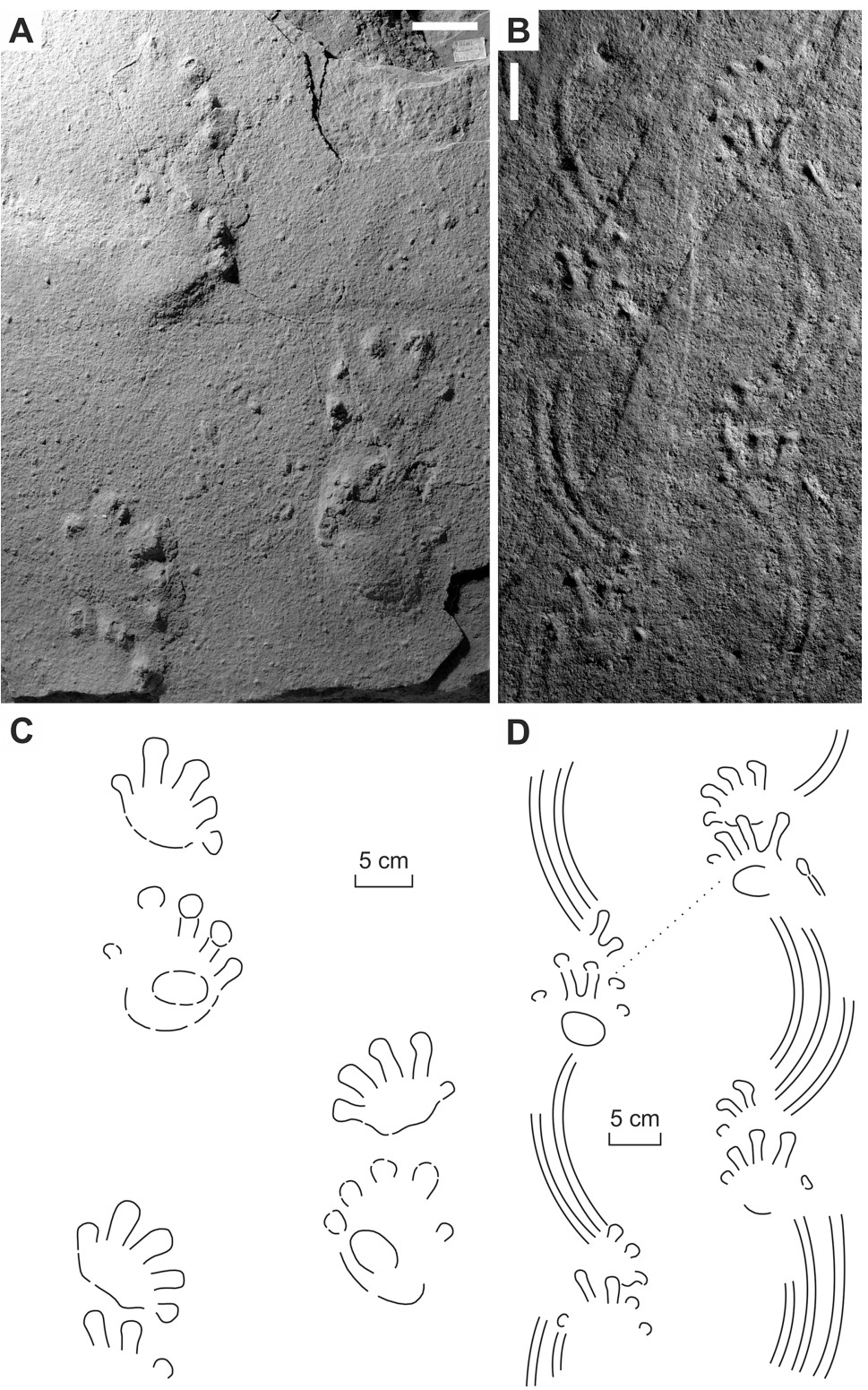

**Figure 5 *Ichniotherium cottae* from the Early Permian Oberhof Formation (Birkheide locality, Thuringian Forest, Germany).** (A, C) MNG 2049; (B, D) NHMS AP-244-19. Unlabeled scale bars equal 5 cm.

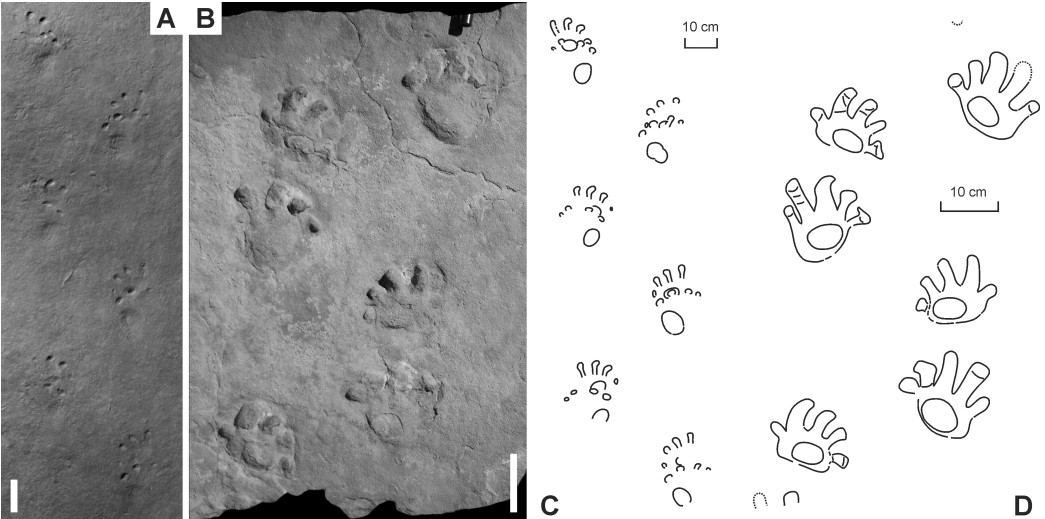

**Figure 6** "*Baropus hainesi*" (*Carman, 1927*) (A, C) and "*Megabaropus hainesi*" (*Baird, 1952*) (B, D) from the Late Carboniferous Monongahela Group (Haine's Farm locality, Ohio, USA). (A, C) OSU 16553; (B, D) CMNH VP-3052. Unlabeled scale bars equal 10 cm.

than those from Birkheide and Gottlob. This pattern is also visible in the bivariate plots $p_{II}(n)$ vs. $p_I(n)$, $p_V(n)$ vs. $p_I(n)$, $m_{II}/m_{IV}$ vs. $m_I/m_{IV}$ and $m_{IV}(n)$ vs. $m_I(n)$ (Figs. 12C–12D). There is also some difference between Birkheide and Gottlob, but usually the degree of overlap is as high as or higher than that between one of them and the Bromacker sample. According to MANOVA the deviation in pedal digit proportions is highly significant ($p < 0.001$) with a highly significant difference in case of the Bromacker and Gottlob samples and a significant difference in case of the Bromacker and Birkheide samples (Table 3). The analysis of manual digit proportions yields a significant difference only between the Bromacker and Birkheide samples; for the combination of $p_I(n)$, $p_{II}(n)$, $m_I(n)$ and $m_{IV}(n)$ test results also indicate a distinction between Bromacker and both of the other sites. In neither pairwise comparison the Birkheide tracks differ significantly from those of Gottlob.

Only one true trackway from Gottlob, consisting of two step cycles, was available for this study. It is considerably closer to the Birkheide sample than to the Bromacker sample in most trackway measures ($\alpha_p$, $\alpha_m$, $P_p(n)$, $C_p(n)$ and $S_p(n)$) whereas the differences in average imprint orientations and trackway width ($B_p(n)$) are small (Table 2; Fig. 13). Comparing the trackway measures for the Birkheide and Bromacker samples, a considerable deviation in manual and pedal pace angulation ($\alpha_p$, $\alpha_m$), pedal pace length ($P_p(n)$) and apparent trunk length ($C(n)$) results in well separated distributions in the plane of greatest variance (PC 1 + PC 2 = 65.2% of variance, Fig. 13A; use of logarithmized length ratios $P_p(n)$, $C_p(n)$ leads only to minor changes in the distribution of groups, see Fig. S2) and significance tests based on all variables or subsets of four or three meaningful variables (Table 4) suggest that this difference between Birkheide and Bromacker is not due to random variation. If the one trackway

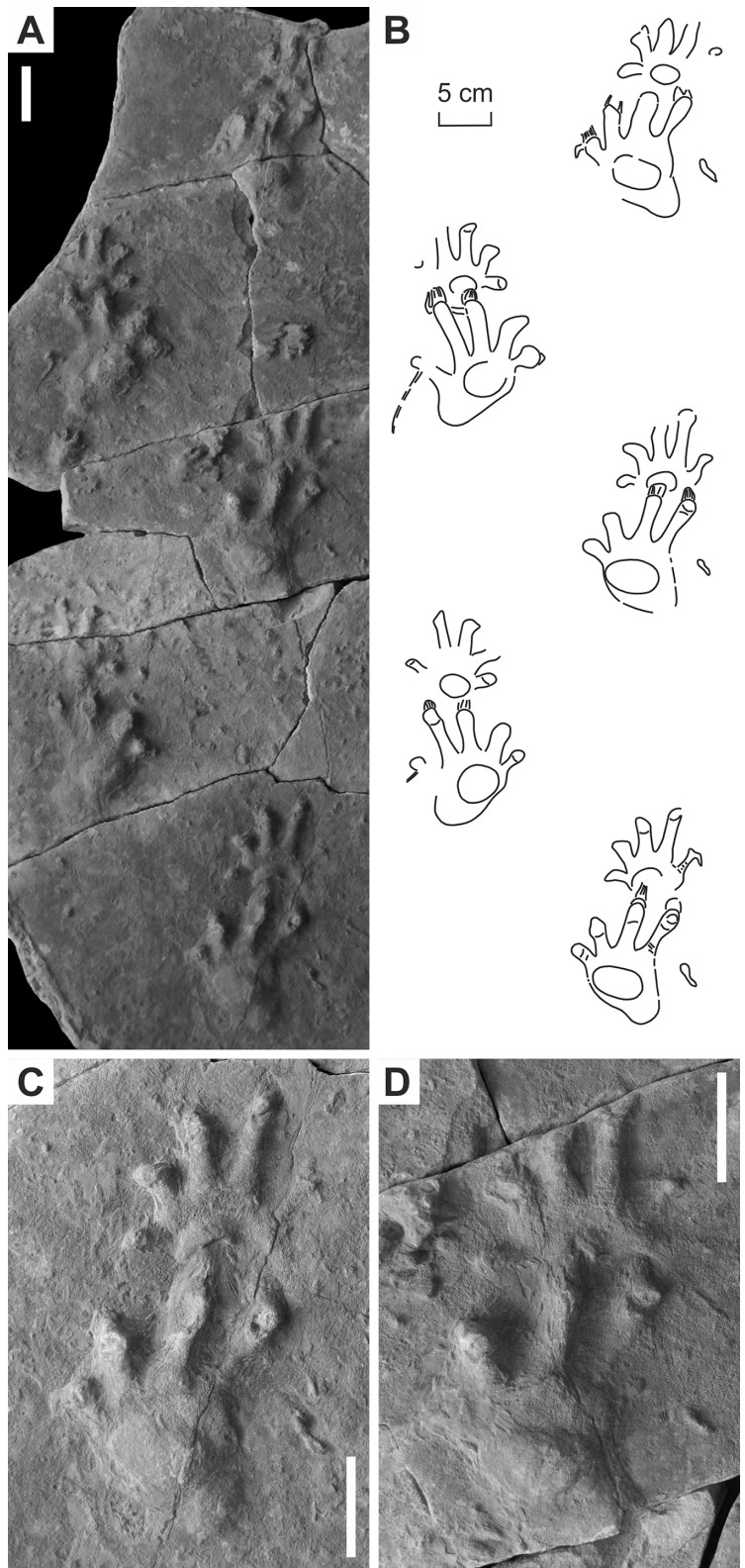

**Figure 7** "*Ichniotherium willsi*" (*Haubold & Sarjeant, 1973*) from the Late Carboniferous Salop Formation (Alveley locality, Birmingham, UK). (A–D) BU 2471. Unlabeled scale bars equal 5 cm.

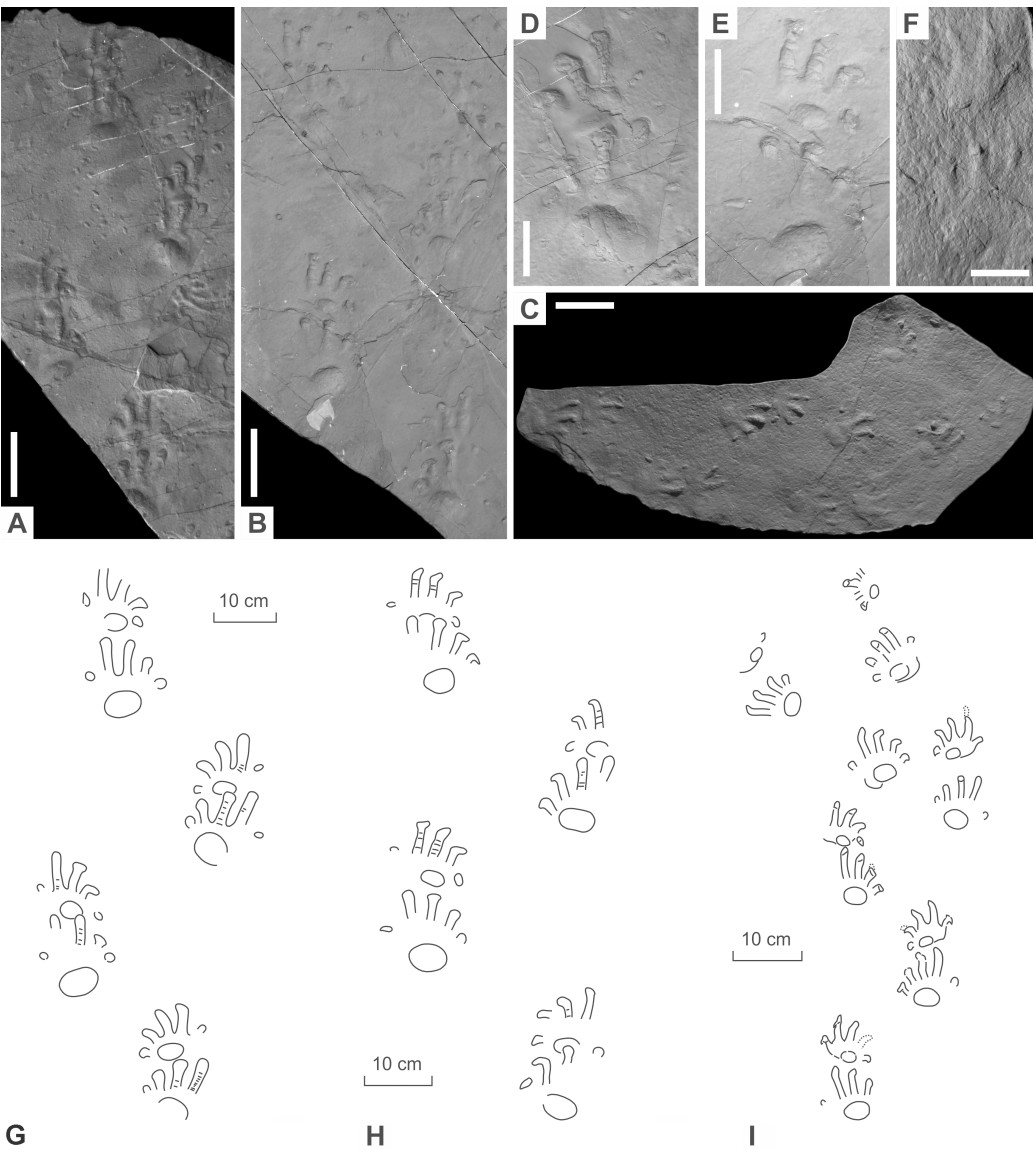

**Figure 8 *Ichniotherium* isp. from the Early Permian Maroon Formation (Maroon localities, Colorado, USA).** (A, D, G) DMNS 50618; (B, E, H) DMNS 50622; (C, F, I) DMNS 55056. Unlabeled scale bars equal 10 cm (A–C) and 5 cm (D–F). 

from Gottlob is added to the data subset from Birkheide, the $p$-values of the employed tests are at least as low as for the Birkheide vs. Bromacker test cases without inclusion of the Gottlob sample (Table 4, last column). Following the lack of separability between the stratigraphically close Gottlob and Birkheide samples, both are considered as a joint sample in the following comparisons.

## Relation of Late Carboniferous and further Early Permian records to the Thuringian sample

Given their relatively high marginal digit lengths ($m_I(n)$, $p_I(n)$, $p_V(n)$), the toe proportions of the Late Carboniferous specimens from Alveley/England and Haine's Farm/Ohio are

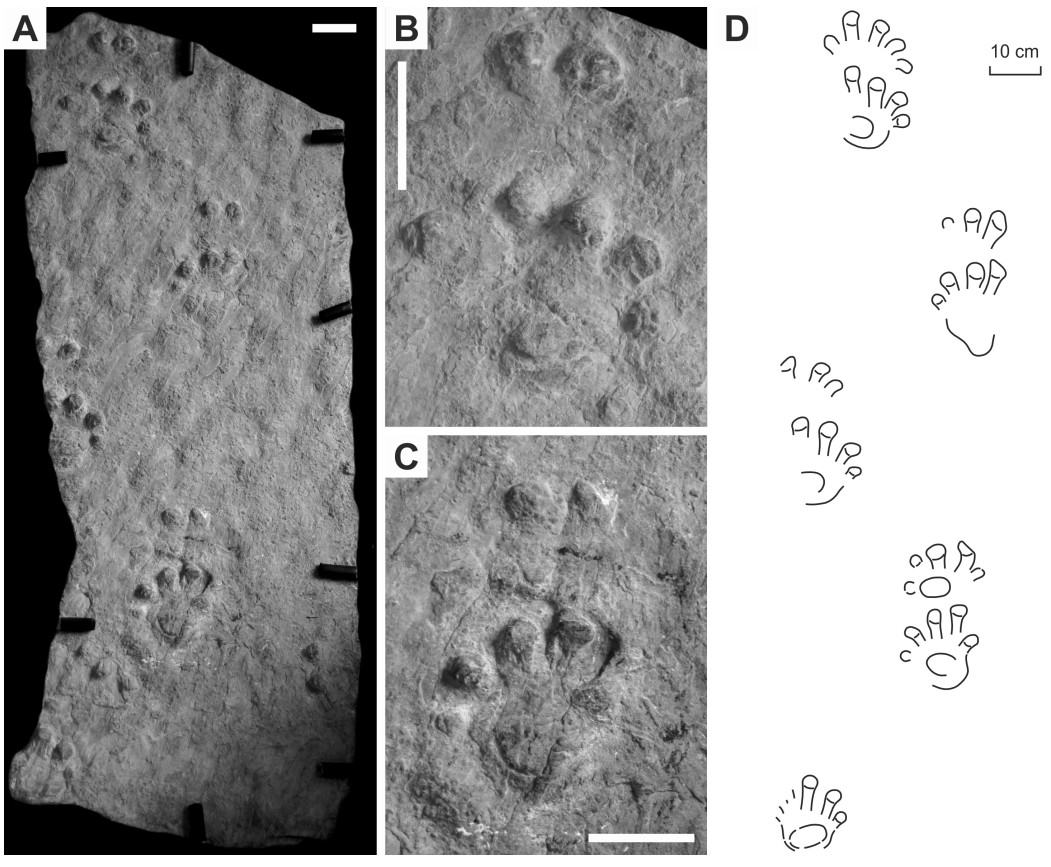

**Figure 9 *Ichniotherium* with relatively short $p_V$ from the Early Permian Washington Formation (Marietta locality, Washington County, Ohio, USA). (A–D) MC-1. Unlabeled scale bars equal 10 cm (A) and 5 cm (B–C).**

usually overlapping considerably with each other and with the distributions of the Birkheide–Gottlob sample but are distinct from that of the Bromacker sample (Figs. 14A–14B). Relative length of the manual digit IV is high for the Alveley trackway (but with a considerable along-track variation between 40 and 54 mm) and causes a certain deviation between Alveley and both, the Haine's Farm and Birkheide–Gottlob samples in the ratio of manual to pedal digit IV ($m_{IV}(n)$; Fig. 14B). Imprint pairs of the Maroon Formation display toe proportions intermediate between the Bromacker sample and the other samples and the best separation from all other samples occurs in the normalized lengths of manual digits I and IV, which are both low in the Maroon Formation record (Fig. 14B). In accordance with the patterns visible in plots of two variables ($p_I(n)$ vs. $p_V(n)$; $m_I(n)$ vs. $m_{IV}(n)$), MANOVA results (Tables 5 and 6) show that the three samples from Birkheide–Gottlob, Haine's Farm and Alveley cannot be distinguished from each other based on toe proportions. The Bromacker imprint pairs differ significantly from those of the other localities with the exception of the Maroon Formation record. If only three groups (Bromacker, Maroon and the rest) and a reduced set of variables ($p_I(n)$, $p_V(n)$, $m_I(n)$, $m_{IV}(n)$) are considered, a moderately exact distinction scheme can be derived (Table 7, see functions F(4) and F(5)).

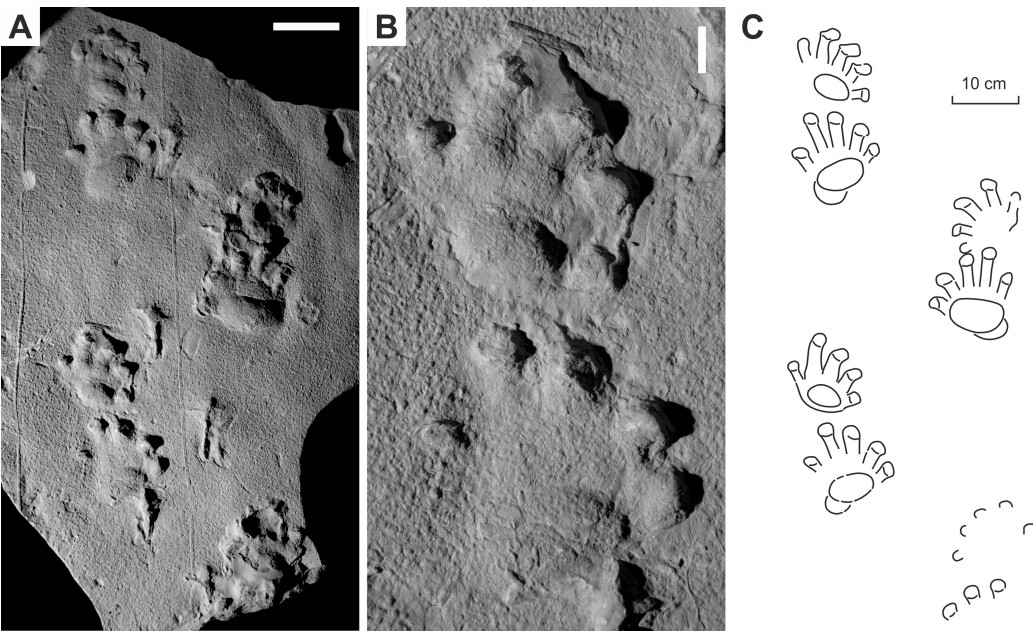

**Figure 10** *Ichniotherium* **with relatively short $p_V$ from the Early Permian Słupiec Formation (Tłumaczow locality, Kłodzko County, Poland).** (A–D) KGM-1. Unlabeled scale bars equal 10 cm (A) and 2 cm (B).

A notably better distinction between the different samples can be reached based on six principal trackway measures ($\beta_p$, $\beta_m$, $C(n)$, $P_p(n)$, $\alpha_m$, $\alpha_p$), which is also visible in a PCA biplot depicting PC 1 and 2 (69.4% of total variance) in which distributions for most localities are only lowly to moderately overlapping (Fig. 14C; a similar result is reached with logarithmized length ratios $C(n)$, $P_p(n)$, see Fig. S2). The Late Carboniferous specimens from Haine's Farm and Alveley differ from other records in their more outward to parallel imprint orientations (Fig. 14D) and display distributions whose centers are often close to each other. The Bromacker sample is marked by comparatively high pace angulations, normalized pace lengths, normalized stride lengths and normalized apparent trunk lengths (Table 2; Figs. 14E and 14F). The more inward orientation of manual and pedal imprints is similar to those of the Birkheide–Gottlob sample and unlike the Late Carboniferous records with the Maroon Formation sample lying in between (Fig. 14D). Because of two step cycles with very low normalized pedal pace length, the distribution of the Maroon Formation sample differs somewhat from the Birkheide–Gottlob and Late Carboniferous samples (Figs. 14E and 14F). According to MANOVA results based on five distinct samples (Bromacker, Birkheide–Gottlob, Haine's Farm, Alveley, Maroon) and six variables ($\beta_p$, $\beta_m$, $C(n)$, $P_p(n)$, $\alpha_m$, $\alpha_p$) only the Bromacker sample differs significantly from the other four (first column in Table 6). When the Late Carboniferous records (Haine's Farm and Alveley) are put in a single group, most of the pairwise test results become significant and only the distinction of the Maroon sample as a group of its own is not well supported (third column in Table 6). Since only three trackway measures

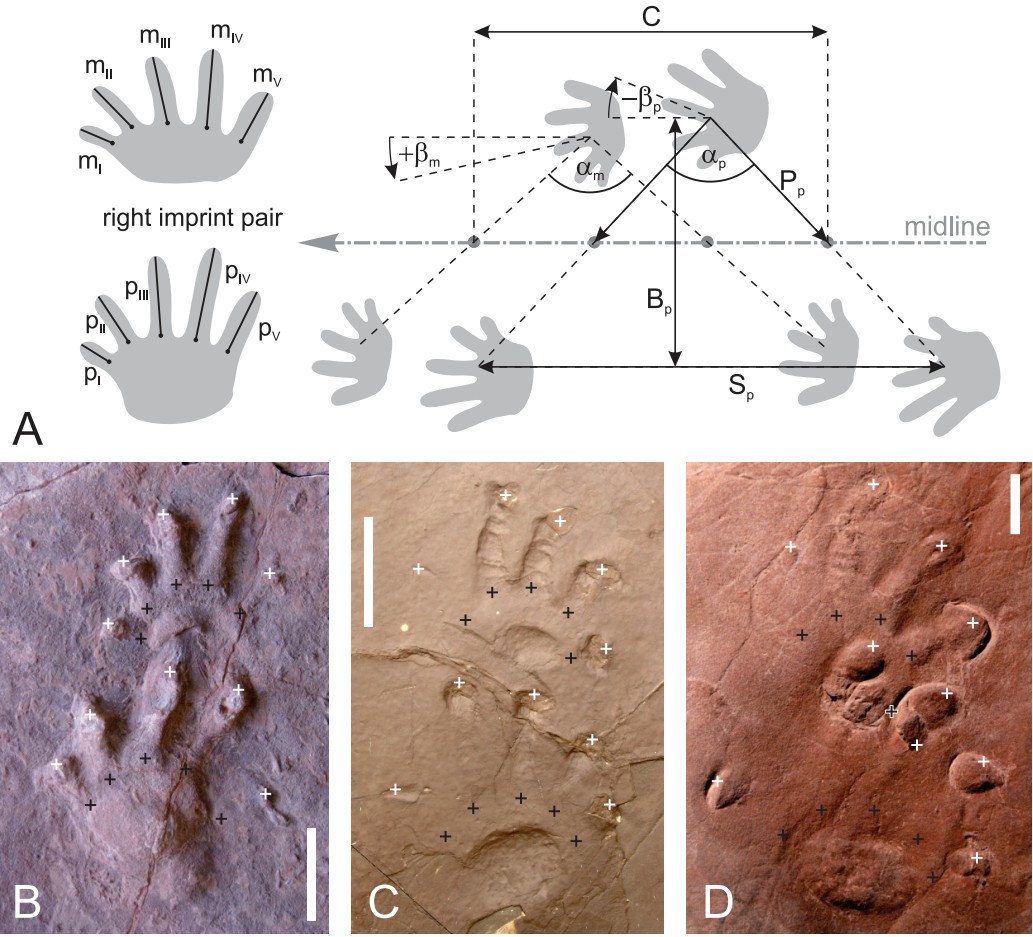

**Figure 11 Imprint and trackway measures used in this approach.** (A) Measures used in this approach include manual and pedal digit lengths ($m_I - p_V$), pace angulation ($\alpha$), imprint orientation ($\beta$), apparent trunk length ($C$), trackway width according to pedal sequence ($B_p$), pedal stride length ($S_p$) and pedal pace length ($P_p$). (B–D) Interpretation of toe tips (white crosses) and toe bases (black crosses) for three imprint pairs that belong to *Ichniotherium* specimens with varying states of preservation: B, BU 2471 (Salop Fm.); (C) DMNS 50622 (Maroon Fm.); (D) MNG 1352 (Tambach Fm.). In case of poorly preserved sole imprints the metering of free digit lengths relies on basal toe points inferred from the orientation of toes, the most clearly preserved toe basis and the outline of the heel pad (which is assumed to be parallel to the trendline connecting all toe bases). In many cases, however, some of the individual toe lengths cannot be determined and have to be considered as missing data.

($\beta_p$, $P_p(n)$, $\alpha_p$) yield most of the variation that is useful for the distinction of the four considered groups (Bromacker, Birkheide–Gottlob, Haine's Farm–Alveley, Maroon) our trackway-pattern-based discrimination scheme is based on this reduced set of variables (columns two and four in Table 6; functions F(1) − F(3) in Table 7, see Figs. 15 and 16).

According to linear discriminant functions based on toe proportions, imprint measures and samples from six localities (Table 7), the two individual trackways from the Słupiec Formation of Tłumaczow/Poland and from the Washington Formation of

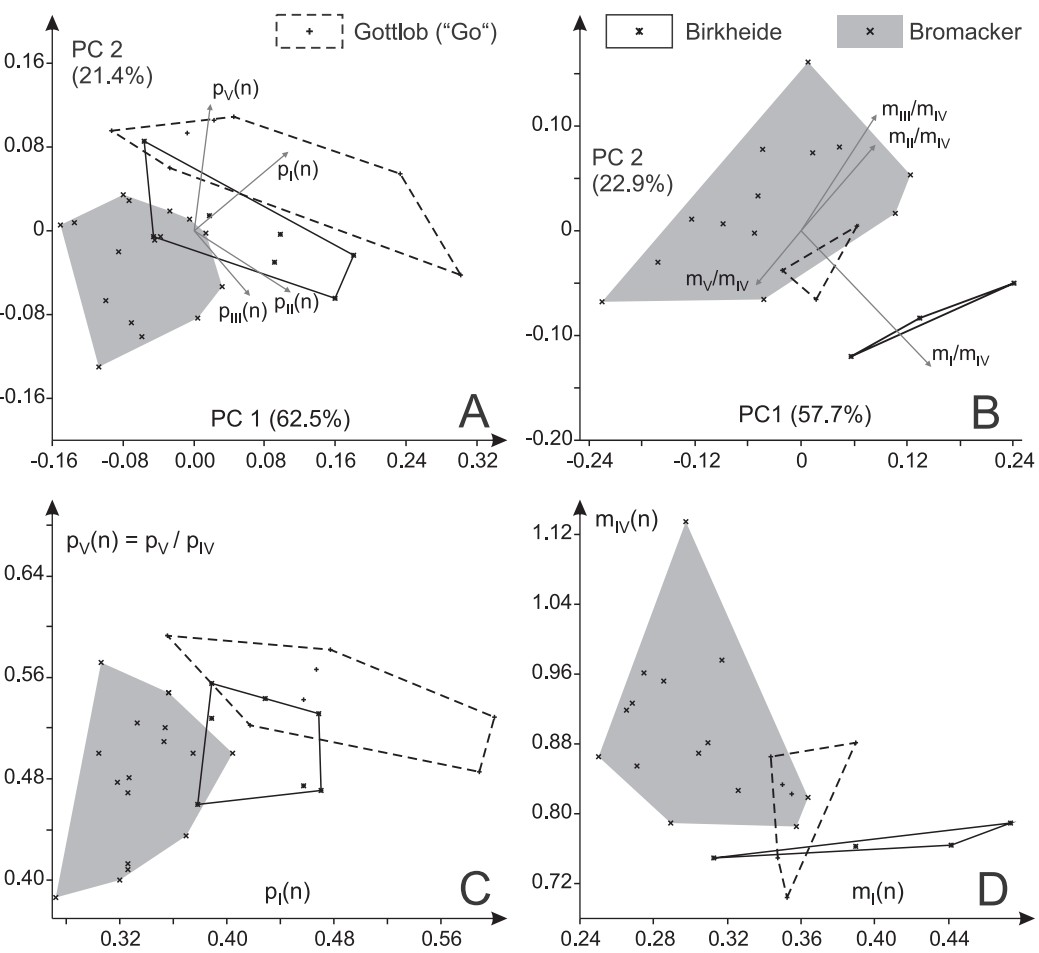

**Figure 12 Variability in the toe ratios of *Ichniotherium cottae* from three Thuringian Forest localities.** (A) Biplot for a principal component analysis (PC 1 vs. 2) based on four manual toe ratios, (B) biplot for a principal component analysis (PC 1 vs. 2) based on four pedal imprint toe ratios. (C, D) Toe ratio plots illustrate notable differences in imprint proportions between different Thuringian Forest localities.               

Marietta/Ohio fall into the spectrum of the Birkheide–Gottlob and Maroon samples (Table 8).

## DISCUSSION

### Homogeneity of the Thuringian Forest sample of *I. cottae*

Based on the present dataset and results at hand we have no sufficient basis for a distinction between the Gottlob and Birkheide samples. The four trackways from Birkheide were made by smaller individuals than those imprints and short series from Gottlob but for the toe proportions the distributions for the two samples either overlap or are close to each other (Tables 2 and 3; Fig. 12).

In the course of this study we found a surprising difference between the Bromacker sample and older trackways from the Birkheide and Gottlob localities when all measures at hand were compared. The Bromacker trackways are marked by high pace angulations, pace lengths, stride lengths and apparent trunk lengths and, apart from

**Table 2 Averages of the normalized digit lengths and trackway measures for all localities.**

*(A) Averages of the normalized digit lengths for all localities, based on imprints with at least four out of five digit lengths and pedal digit length IV measurably preserved*

| Locality | $p_I(n)$ | $p_{II}(n)$ | $p_{III}(n)$ | $p_V(n)$ | $m_I(n)$ | $m_{II}(n)$ | $m_{III}(n)$ | $m_{IV}(n)$ | $m_V(n)$ |
|---|---|---|---|---|---|---|---|---|---|
| Gottlob | 0.506 | 0.651 | 0.833 | 0.540 | 0.356 | 0.521 | 0.681 | 0.829 | 0.423 |
| Birkheide | 0.416 | 0.651 | 0.836 | 0.506 | 0.375 | 0.503 | 0.638 | 0.766 | 0.371 |
| Bromacker | 0.350 | 0.614 | 0.814 | 0.473 | 0.285 | 0.586 | 0.753 | 0.910 | 0.461 |
| Haine's Farm | 0.493 | 0.646 | 0.892 | 0.536 | 0.366 | 0.528 | 0.689 | 0.774 | 0.416 |
| Alveley | 0.437 | 0.620 | 0.823 | 0.570 | 0.372 | 0.498 | 0.660 | 0.864 | 0.454 |
| Maroon (two individual sites) | 0.389 | 0.585 | 0.816 | 0.558 | 0.295 | 0.475 | 0.653 | 0.783 | 0.404 |
| Marietta | 0.437 | 0.669 | 0.822 | 0.488 | 0.356 | 0.500 | 0.647 | 0.825 | 0.471 |
| Tłumaczow | 0.402 | 0.584 | 0.831 | 0.589 | 0.298 | 0.543 | 0.684 | 0.856 | 0.498 |

*(B) Averages of trackway measures for all localities*

| Locality | $\alpha_p$ | $\alpha_m$ | $\beta_p$ | $\beta_m$ | $P_p(n)$ | $C(n)$ | $S_p(n)$ | $B_p(n)$ |
|---|---|---|---|---|---|---|---|---|
| Gottlob | 85.0 | 84.0 | 12.5 | 33.0 | 4.24 | 4.17 | 5.61 | 3.16 |
| Birkheide | 94.5 | 93.3 | 8.4 | 23.8 | 4.68 | 4.73 | 6.45 | 3.29 |
| Bromacker | 105.5 | 109.2 | 10.1 | 24.8 | 5.18 | 5.44 | 8.10 | 3.12 |
| Haine's Farm | 89.6 | 91.9 | −27.9 | 11.7 | 4.70 | 4.36 | 6.48 | 3.37 |
| Alveley | 102.3 | 92.0 | −13.5 | 6.3 | 4.67 | 5.26 | 7.27 | 2.92 |
| Maroon (two sites) | 97.8 | 92.0 | −3.7 | 12.0 | 4.08 | 4.56 | 6.15 | 2.69 |
| Marietta | 107.7 | 102.0 | 3.0 | 9.0 | 4.53 | 5.06 | 7.38 | 2.64 |
| Tłumaczow | 106.0 | 90.0 | 1.0 | 34.5 | 4.29 | 5.12 | 6.80 | 2.63 |

**Table 3 Multivariate analysis of variance (MANOVA) results documenting the distinctiveness of Thuringian Forest records according to different sets of toe ratios.**

| Set of variables | Wilk's lambda test (p-value) | Hoteling test (Bonferroni-corrected p-values) | | |
|---|---|---|---|---|
| | | Brom vs. Gott | Brom vs. Birk | Gott vs. Birk |
| $p_I(n)$ to $p_V(n)$ Sample size: 27 | $4.579 \times 10^{-6}$ | $8.27921 \times 10^{-5}$ | $0.0232219$ | 0.109217 |
| $m_I/m_{IV}$ to $m_V/m_{IV}$ Sample size: 17 | $0.004365$ | 0.588676 | $0.00295647$ | 1 |
| $p_I(n)$, $p_{II}(n)$, $m_I(n)$, $m_{IV}(n)$ Sample: 22 | $3.064 \times 10^{-6}$ | $0.00995011$ | $0.001823$ | 0.14639 |

Notes:
  Null hypothesis that samples come from the same statistic population is not declined if *p*-value > 0.05. Numbers printed in italics represent significant test results.

that, some of the marginal toes ($m_I$, $p_I$, $p_V$) were conspicuously shorter than in the older Thuringian Forest tracks (Figs. 12 and 13, Tables 3 and 4). Moreover, in the Bromacker step cycles low pedal pace angulation appears to be compensated by high normalized pace length and vice versa ($r = −0.419$, see Fig. 13F)—at the benefit of normalized stride length that does not fall below a certain value (6.25). Taken together, these differences can arguably not be attributed to substrate differences or allometry in a functionally and taxonomically identical trackmaker (given the similarly small imprint sizes in both, Birkheide and Bromacker tracks) but actually reflect functionally distinct trackmakers. One step cycle of a Bromacker trackway, SSB-1, with relatively short pace length causes much of the overlap with the Birkheide and

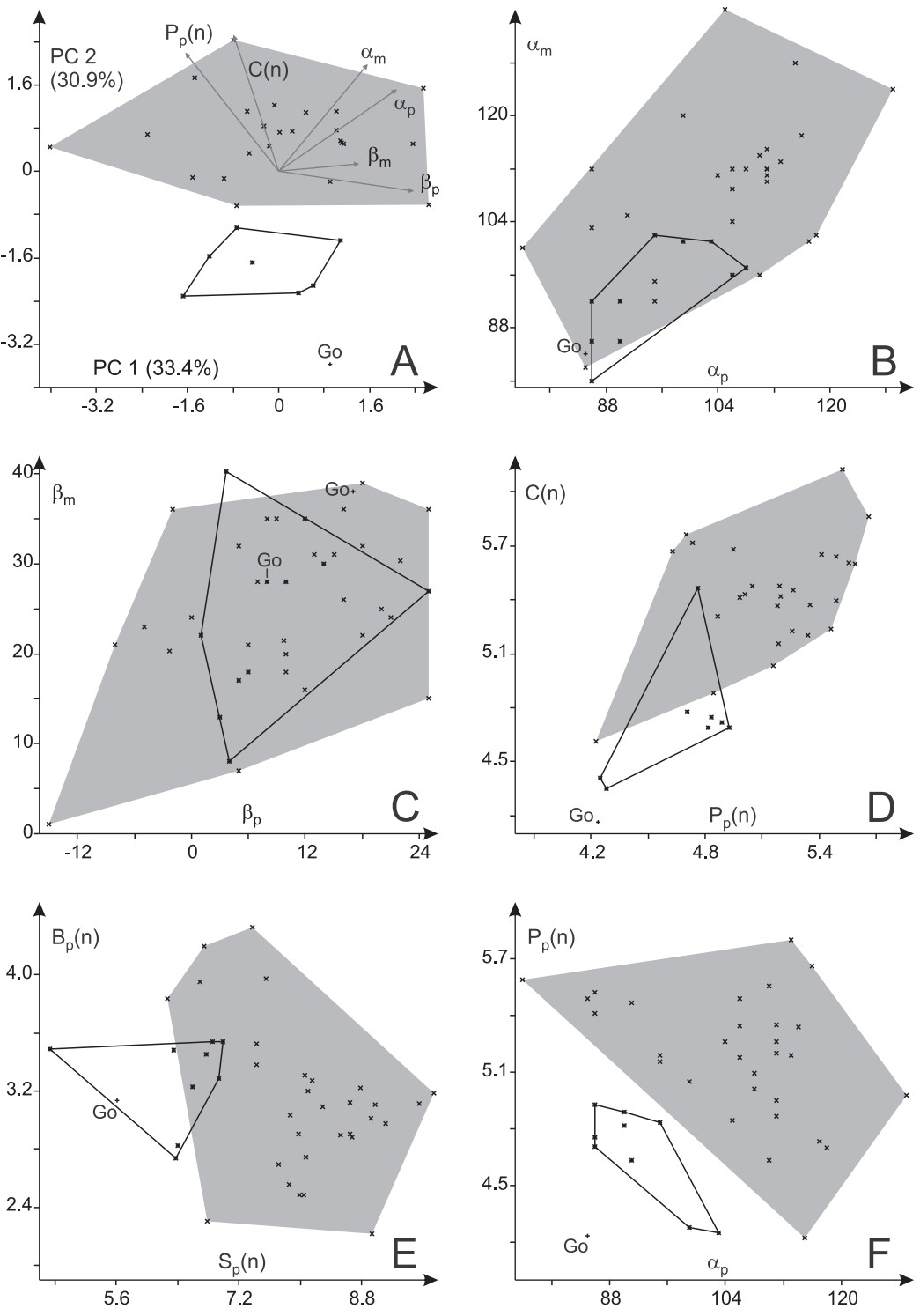

**Figure 13 Variability in trackway parameters of *Ichniotherium cottae* from three Thuringian Forest localities.** (A) Biplot for a principal component analysis (PC 1 vs. 2) based on six variables. (B) Plot of manual vs. pedal pace angulation, (C) manual vs. pedal imprint orientation; (D) apparent trunk length (normalized) vs. pedal pace length (normalized), (E) gauge width (normalized) vs. stride length (normalized), (F) pedal pace length (normalized) vs. pedal pace angulation. Go—step cycles from Gottlob quarry.

**Table 4 Multivariate analysis of variance (MANOVA) results documenting the distinctiveness of Thuringian Forest records.**

| Set of variables | Bromacker vs. Birkheide (Hoteling test $p$-value) | Bromacker vs. Birkheide + Gottlob (Hoteling test $p$-value) |
|---|---|---|
| $\beta_p$, $\beta_m$, $C(n)$, $P_p(n)$, $\alpha_m$, $\alpha_p$ Sample: 29/30 | $3.5855 \times 10^{-6}$ | $1.90152 \times 10^{-6}$ |
| $C(n)$, $P_p(n)$, $\alpha_m$, $\alpha_p$ Sample size: 33/34 | $5.96582 \times 10^{-9}$ | $3.00141 \times 10^{-9}$ |
| $\beta_p$, $\beta_m$, $C(n)$, $P_p(n)$, $\alpha_m$, $\alpha_p$ Sample size: 31/32 | $0.04437$ | $0.0181592$ |

**Notes:**
Different sets of trackway parameters are tested (null hypothesis that samples come from the same statistic population is not declined if $p$-value $> 0.05$). Numbers printed in italics represent significant test results.

Gottlob samples (Figs. 13D and 13E, dashed area in Fig. 15A). However, measures defining the trackway pattern suggest that this individual step cycle might represent a part of a curved path and accordingly differs from the rest of the Bromacker sample (see Figs. 4C and 4I).

## Non-Thuringian record and distinction of trackway morphotypes

According to toe proportions and most trackway measures the three trackways from the Late Carboniferous of Alveley/Great Britain and Haine's Farm/Ohio fall within the range of the Thuringian Forest sample—with one notable exception: Their pedal imprints share a distinctive outward rotation (with respect to the direction of movement), a feature also noted in earlier discussions of the Alveley specimen ("*Ichniotherium willsi*," see *Voigt & Ganzelewski, 2010*), and the manual imprints often display a more parallel-to-midline orientation ($<18°$ inward rotation) than those of the Thuringian Forest specimens ($\beta_p$ and $\beta_m$ angles; see Figs. 14D and 15A). Whereas length ratios and pace angulations might be more substrate-dependant, we consider imprint orientation as one of the trackway pattern characteristics that is likely to be anatomically controlled and indicative of a functionally distinct trackmaker. The separability of the combined Alveley and Haine's Farm sample from the Bromacker and Birkheide–Gottlob samples is supported by trackway-parameter-based multivariate analyses of variance (columns 3 and 4 in Table 6).

Apart from very low normalized pace length values in two step cycles (of specimen DMNS 55056, Figs. 8C, 8J, 14F and 15) and subtle differences in the toe proportions of the manual imprints (relatively short manual digit IV, see Fig. 12B), the sample of three trackways from the Maroon Formation falls into the ranges of the previously distinguished groups (Birkheide–Gottlob, Bromacker, Alveley + Haine's Farm). Their trackway measures are mostly overlapping with the Birkheide–Gottlob sample but they show more parallel imprint orientations that correspond to those seen in some step cycles of the Bromacker sample (Fig. 14D). Even though some MANOVA results support a distinction of the Maroon Formation tracks at higher significance levels (0.05, 0.01; see Table 6) it would fail at lower significance levels (0.001, 0.0001) and we find it likely that the deviations in length proportions of one Maroon trackway are not anatomically controlled. Furthermore, our step-cycle-based analyses have the caveat that step cycles from the same trackway can hardly be regarded as independent observations, a requirement of the applied statistic tests which is not fully met in our approach (therefore

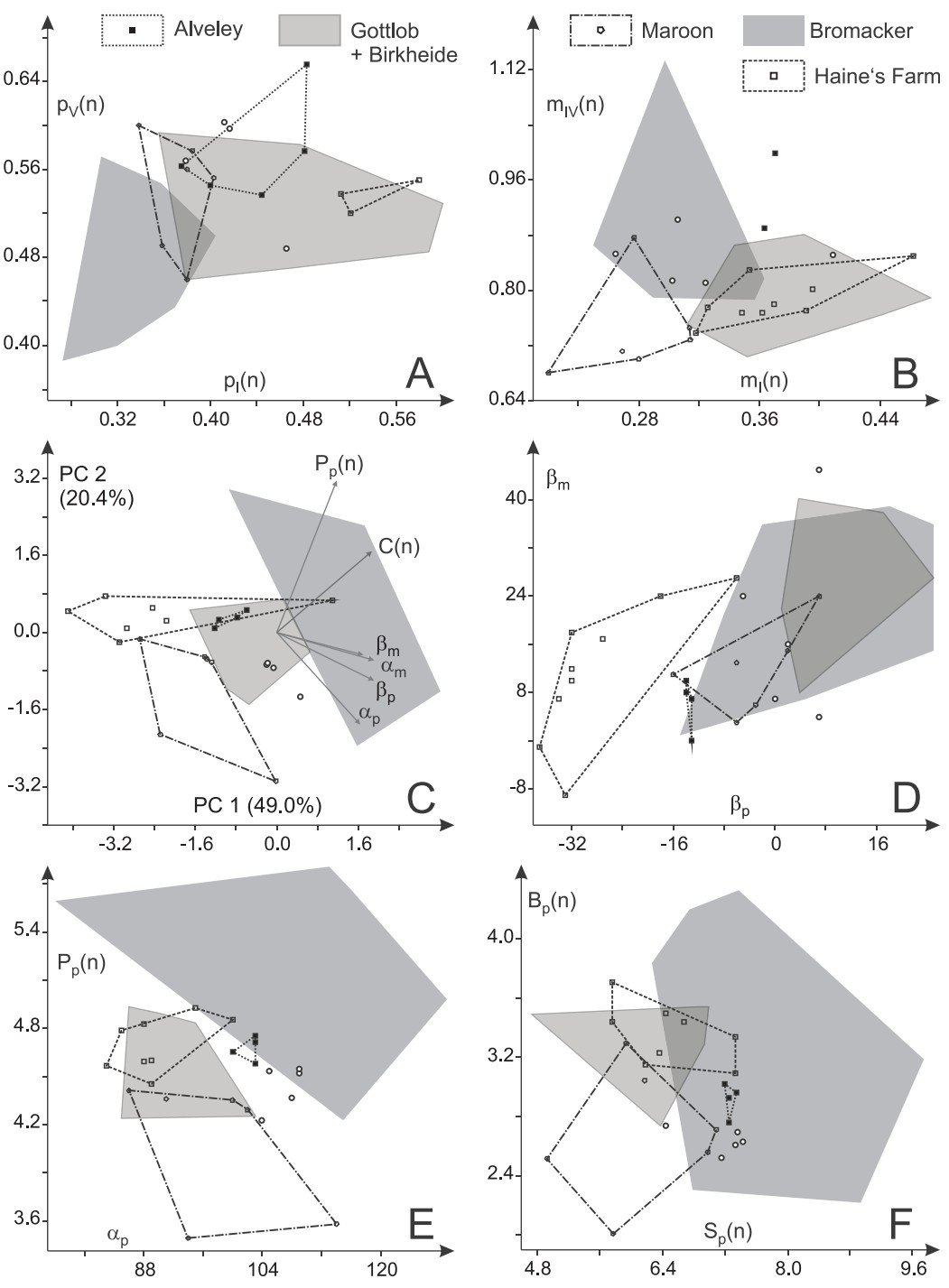

**Figure 14 Variability in toe proportions and trackway parameters of *Ichniotherium cottae* with relatively short pedal digit V from nine localities.** (A, B) Toe ratio plots; (C) biplot for a principal component analysis (PC 1 vs. 2) based on six variables; (D) manual vs. pedal imprint orientation; (E) pedal pace length (normalized) vs. pedal pace angulation; (F) gauge width (normalized) vs. stride length (normalized). Unfilled circles represent average toe ratios and step cycles of the Tłumaczow and Marietta specimens.

**Table 5 Multivariate analysis of variance (MANOVA) results documenting the distinctiveness of *Ichniotherium* samples from six localities.**

| Test cases | $p_I(n)$ to $p_V(n)$ Sample: 41 | $m_I/m_{IV}$ to $m_V/m_{IV}$ Sample: 25 | $p_I(n), p_{II}(n), m_I(n), m_{IV}(n)$ Sample: 40 | $p_I(n) \dots m_{IV}(n)$ Hain incl. Alve Sample: 40 |
|---|---|---|---|---|
| Total (*p*-value for Wilk's lambda) | $1.277 \times 10^{-6}$ | 0.01946 | $8.252 \times 10^{-9}$ | $1.425 \times 10^{-8}$ |
| Brom vs. Birk/Gott | 0.000472484 | 0.0417876 | 0.00123679 | 0.00101849 |
| Brom vs. Hain | 0.00862805 | 0.344206 | 0.00436374 | 0.00319036 |
| Brom vs. Alve | 0.0596709 | – | 1 | – |
| Brom vs. Maro | 1 | 1 | 0.0475402 | 0.0585216 |
| Birk/Gott vs. Hain | 0.405526 | 1 | 1 | 1 |
| Birk/Gott vs. Alve | 1 | – | 1 | – |
| Birk/Gott vs. Maro | 0.821683 | 1 | 0.0031644 | 0.00169635 |
| Hain vs. Alve | 1 | – | 1 | – |
| Hain vs. Maro | 0.920184 | 1 | 0.0111189 | 0.00155712 |
| Alve vs. Maro | 1 | – | 0.729014 | – |

**Notes:**
Different sets of toe proportions are tested (null hypothesis that samples come from the same statistic population is not declined if *p*-value > 0.05). For pairwise comparisons the Bonferroni-corrected *p*-values of Hotelling tests are listed. Numbers printed in italics represent significant test results.

**Table 6 Multivariate analysis of variance (MANOVA) results documenting the distinctiveness of *Ichniotherium* samples from six localities.**

| Test cases | Alveley as separate sample | | Haine's Farm incl. Alveley | |
|---|---|---|---|---|
| | $\beta_p, \beta_m, C(n), P_p(n), \alpha_p, \alpha_m$ Sample: 49 | $\alpha_p, \beta_p, P_p(n)$ Sample: 52 | $\beta_p, \beta_m, C(n), P_p(n), \alpha_p, \alpha_m$ Sample: 47 | $\alpha_p, \beta_p, P_p(n)$ Sample: 52 |
| Total (*p*-value for Wilk's lambda) | $1.631 \times 10^{-17}$ | $2.708 \times 10^{-22}$ | $1.088 \times 10^{-16}$ | $8.345 \times 10^{-23}$ |
| Brom vs. Birk/Gott | 0.00022799 | $1.35 \times 10^{-6}$ | 0.000256636 | $8.14 \times 10^{-7}$ |
| Brom vs. Hain | $3.62 \times 10^{-7}$ | $4.35 \times 10^{-10}$ | $3.91 \times 10^{-8}$ | $4.62 \times 10^{-11}$ |
| Brom vs. Alve | 0.00262849 | 0.00024067 | – | – |
| Brom vs. Maro | $5.57 \times 10^{-6}$ | $1.89 \times 10^{-8}$ | $4.38 \times 10^{-6}$ | $1.42 \times 10^{-8}$ |
| Birk/Gott vs. Hain | 0.0413744 | 0.000105081 | 0.00207178 | $8.04 \times 10^{-6}$ |
| Birk/Gott vs. Alve | 0.552279 | 0.0315482 | – | – |
| Birk/Gott vs. Maro | 0.774264 | 0.0421465 | 0.441969 | 0.0264885 |
| Hain vs. Alve | 1 | 1 | – | – |
| Hain vs. Maro | 0.815793 | 0.0248551 | 0.173938 | 0.0048175 |
| Alve vs. Maro | 1 | 0.554441 | – | – |

**Notes:**
Different sets of trackway parameters are tested (null hypothesis that samples come from the same statistic population is not declined if *p*-value > 0.05). For pairwise comparisons the Bonferroni-corrected *p*-values of Hotelling tests are listed. Numbers printed in italics represent significant test results.

application of lower significance levels). In sum we regard only three morphotypes of *Ichniotherium* with relatively short pedal digit V as sufficiently supported by the present dataset: the "Birkheide–Gottlob type" with is based on trackways and short series from the Gottlob and Birkheide localities, the "Bromacker type" which based on trackways from the Bromacker locality and the "Hainesi–Willsi type" which is based on trackways from the Alveley and Haine's Farm localities. Pending further observations, the Maroon sample can be tentatively referred to the morphologically similar Birkheide–Gottlob type.

**Table 7 List of suggested discrimination criteria, each based on two variables (plus length of pedal digit IV).**

| Discrimination steps | Linear discriminant function | Hotelling test |
|---|---|---|
| (1) Brom vs. all others | $F(1) = 0.226 \times \alpha_p + 7.786 \times P_p(n) - 60.405$ | $p = 8.024 \times 10^{-14}$ |
| (2) Hain/Alve vs. Birk/Gott/Maro | $F(2) = 0.30594 \times \beta_p - 5.4972 \times P_p(n) + 27.8749$ | $p = 1.718 \times 10^{-7}$ |
| (3) Birk/Gott vs. Maro | $F(3) = 0.26379 \times \beta_p + 5.3169 \times P_p(n) - 24.0727$ | $p = 0.002283$ |
| (4) Maro vs. all others | $F(4) = 20.402 \times m_{IV}(n) + 35.512 \times m_I(n) - 27.1769$ | $p = 0.000195$ |
| (5) Birk/Gott/Hain/Alve vs. Brom | $F(5) = 36.926 \times p_I(n) + 22.899 \times p_V(n) - 26.3278$ | $p = 2.119 \times 10^{-7}$ |

Notes:
Angles ($\alpha$, $\beta$) shall be included with unit "degree." Numbers printed in italics represent significant test results.

According to a set of linear discriminant functions derived from more specimen-rich samples, the trackways from Tłumaczow and Marietta are grouping with the Birkheide–Gottlob and Maroon samples (Tables 7 and 8). Thus, we tentatively refer them to the Birkheide–Gottlob type here as well.

## Ichnotaxonomic consequences

*Voigt (2005)* and *Voigt, Berman & Henrici (2007)* distinguished two ichnospecies for *Ichniotherium* based on the co-occurrence of two diadectids and two corresponding morphologically distinct types of reptiliomorph footprints at the Bromacker locality. By including *Schmidtopus praesidentis* (*Schmidt, 1956*), an over 310-million-year old trackway of a large diadectomorph or possibly a more basal member of the amniote stem group, *Voigt & Ganzelewski (2010)* expanded the morphological and temporal range of *Ichniotherium* considerably. Notwithstanding the problematic status of *I. praesidentis* (that shall be discussed elsewhere), we keep the taxonomic scheme of an ichnogenus *Ichniotherium* with several species that shall at least include *I. sphaerodactylum* with relatively long pedal digit V and all tracks considered here with relatively short pedal digit V. The first available binomial name for *Ichniotherium* tracks with relatively short pedal digit V is *I. cottae* (*Pohlig, 1885*) which is redefined here as all "*Ichniotherium* with relatively short pedal digit V (i.e., pedal digit ratio V/IV < 0.6)." Based on distinct trackway patterns and the somewhat weaker signal of variation in toe proportions we propose three morphotypes of *I. cottae* (Fig. 15B):

**Birkheide–Gottlob type.** Referred specimens: HF 57, HF 89, MNG 1381, MNG 1382, MNG 1385, MNG 1387, MNG 1386-F1, MNG 1781 from the *I. cottae* type locality Gottlob Quarry/Friedrichroda and MNG 2047, MNG 2049, NHMS AP-244-19, NHMS P-418 from Birkheide Quarry/Tambach-Dietharz.

Diagnosis: *Ichniotherium* with ratio $p_V/p_{IV} < 0.6$, parallel to inward rotation of the pedal imprints (1°–25°) and manual imprints (8°–40°), pace angulations: 80°–102° (manual), 85°–108° (pedal), pedal pace length/$p_{IV}$: 4.2–5.0, apparent trunk length/$p_{IV}$: 4.1–5.5, pedal stride length/$p_{IV}$: 4.7–7.0, gauge width (pedal)/$p_{IV}$: 2.7–3.6. Toe ratios based on imprints with at least four digit lengths preserved: $p_I/p_{IV}$: 0.36–0.60, $p_V/p_{IV}$: 0.46–0.59, $m_I/p_{IV}$: 0.31–0.47, $m_{IV}/p_{IV}$: 0.70–0.88.

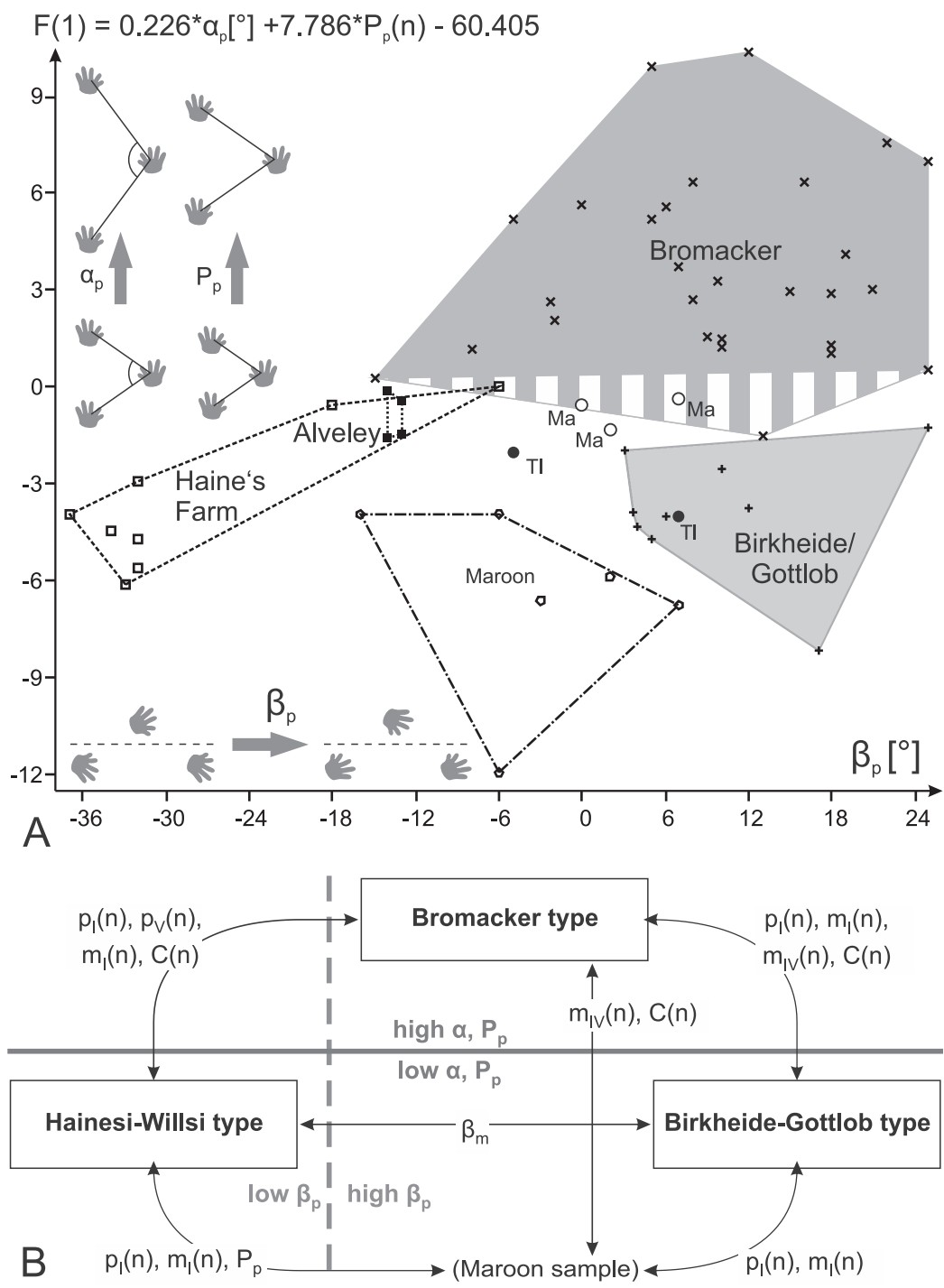

**Figure 15 Morphospace occupation by different samples and morphotypes of *Ichniotherium* with relatively short pedal digit V.** (A) Plot of linear discriminant function values against angle of pedal imprint orientation ($\beta_p$); (B) scheme illustrating trackway parameters and toe ratios that allow pairwise distinctions between three morphotypes and the Maroon sample (two localities). Labels: Ma, Marietta/Ohio; Tl, Tłumaczow/Poland.

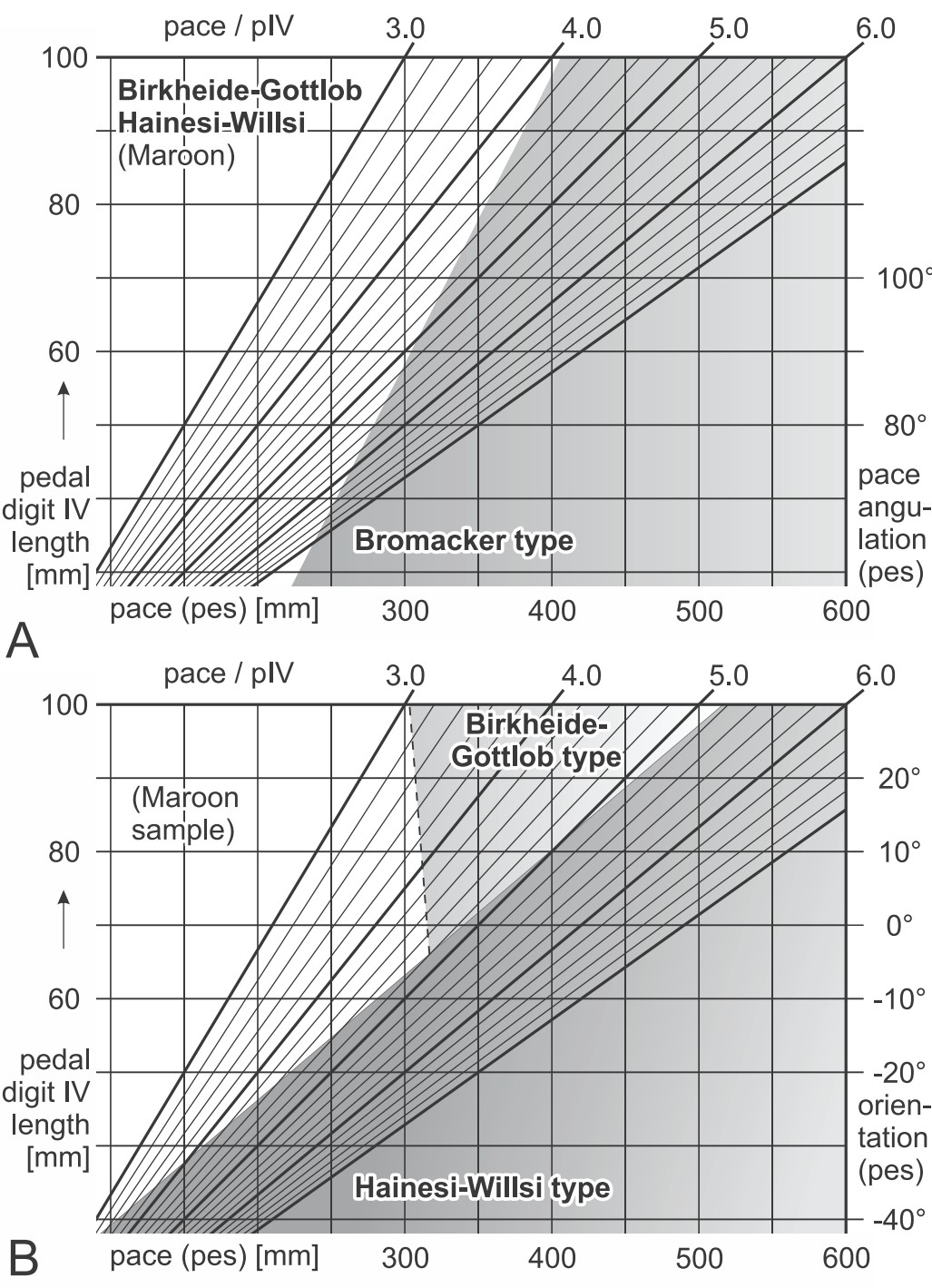

**Figure 16 Nomograms for simple discrimination of *Ichniotherium cottae* morphotypes in the field.**

Differential diagnosis: Negative values for $F(1) = 0.226 \times \alpha_p + 7.786 \times P_p(n) - 60.405$, positive values for $F(2) = 0.30594 \times \beta_p - 5.4972 \times P_p(n) + 27.8749$; positive values for $F(5) = 36.926 \times p_I(n) + 22.899 \times p_V(n) - 26.3278$ (see Table 7; Fig. 16). Linear

**Table 8 Discriminant function scores and classification of trackways from Tłumaczow/Poland and Marietta/Ohio.**

| Specimen | Step cycle | $\alpha_p$ | $P_p(n)$ | $\beta_p$ | F(1) | F(2) | F(3) | Result |
|---|---|---|---|---|---|---|---|---|
| MC-1 | 2 | 105 | 4.53 | 2 | −1.38 | 3.56 | 0.56 | B/G |
| MC-1 | 3 | 109 | 4.52 | 0 | −0.60 | 3.04 | −0.06 | Maroon |
| KGM-1 | 1 | 108 | 4.36 | −5 | −2.04 | 2.37 | −2.20 | Maroon |
| KGM-1 | 2 | 104 | 4.22 | 7 | −4.03 | 6.81 | 0.22 | B/G |

| Specimen | Couple | $p_I$ | $p_V$ | $m_I$ | $m_{IV}$ | F(4) | F(5) | Result |
|---|---|---|---|---|---|---|---|---|
| MC-1 | 2 | 0.47 | 0.49 | 0.30 | 0.81 | 0.17 | 2.03 | B/G + H/W |
| KGM-1 | 2 | 0.42 | 0.60 | 0.31 | 0.90 | 2.09 | 2.73 | B/G + H/W |
| KGM-1 | 3 | 0.41 | 0.60 | 0.26 | 0.85 | −0.37 | 2.68 | Maroon |
| KGM-1 | 4 | 0.38 | 0.57 | 0.32 | 0.81 | 0.88 | 0.64 | B/G + H/W |

discriminant functions based on toe proportions allow no separation from the Hainesi–Willsi type.

**Bromacker type.** Referred specimens: MB.ICV.3-F1, MB.ICV.3-F2, MNG 1352, MNG 1819, MNG 2356-16-F1, MNG 2356-16-F2, MNG 10179, MSEO-I-36, PMJ P-1321-F3, SSB-1, UGKU 130-F1, UGKU 130-F2.

Diagnosis: *Ichniotherium* with ratio $p_V/p_{IV} < 0.6$, usually parallel to inward rotation of the pedal imprints (−15° to 25°) and manual imprints (1°–39°), pace angulations: 82°–136° (manual), 76°–129° (pedal), pedal pace length/$p_{IV}$: 4.2–5.8, apparent trunk length/$p_{IV}$: 4.6–6.2, pedal stride length/$p_{IV}$: 6.2–9.8, gauge width (pedal)/$p_{IV}$: 2.2–4.4. Toe ratios based on imprints with at least four digit lengths preserved: $p_I/p_{IV}$: 0.27–0.48, $p_V/p_{IV}$: 0.39–0.55, $m_I/p_{IV}$: 0.25–0.36, $m_{IV}/p_{IV}$: 0.79–1.14.

Differential diagnosis: Positive values for F(1) = 0.226 × $\alpha_p$ + 7.786 × $P_p(n)$ − 60.405; negative values for F(5) = 36.926 × $p_I(n)$ + 22.899 × $p_V(n)$ − 26.3278.

**Hainesi–Willsi type.** Referred specimens: OSU 16553, CMNH VP-3052 from Haine's Farm/Ohio and BU 2471 from Alveley/Great Britain.

Diagnosis: *Ichniotherium* with ratio $p_V/p_{IV} < 0.6$ (trackway average), usually outward rotation of pedal imprints (−37° to −6°) and parallel orientation of manual imprints (−9° to 27°), pace angulations: 82°–108° (manual), 83°–103° (pedal), pedal pace length/$p_{IV}$: 4.4–5.0, apparent trunk length/$p_{IV}$: 4.0–6.0, pedal stride length/$p_{IV}$: 5.7–7.4, gauge width (pedal)/$p_{IV}$: 2.7–3.8. Toe ratios based on imprints with at least four digit lengths preserved: $p_I/p_{IV}$: 0.38–0.58, $p_V/p_{IV}$: 0.52–0.66, $m_I/p_{IV}$: 0.32–0.46, $m_{IV}/p_{IV}$: 0.74–1.

Differential diagnosis: Negative values for F(1) = 0.226 × $\alpha_p$ + 7.786 × $P_p(n)$ − 60.405, negative values for F(2) = 0.30594 × $\beta_p$ − 5.4972 × $P_p(n)$ + 27.8749; positive values for F(5) = 36.926 × $p_I(n)$ + 22.899 × $p_V(n)$ − 26.3278. Linear discriminant functions based on toe proportions allow no separation from the Birkheide–Gottlob type.

Not considered as a type of its own here, the sample of three trackways from the Maroon Formation (DMNS 50618, DMNS 50622, DMNS 55056) can be distinguished from the three morphotypes by a combination of low normalized pace lengths and

pedal imprint orientations that are rather parallel to the trackway midline (negative values for F(3) = $0.26379 \times \beta_p + 5.3169 \times P_p(n) - 24.0727$, see Table 7; Fig. 16). According to the linear discriminant functions F(1) and F(2) the Maroon trackways and those from Tłumaczow/Poland and Marietta/Oklahoma are classified as Birkheide–Gottlob type.

As discussed above, neither the samples from Gottlob and Birkheide nor those from Haine's Farm and Alveley display a complete overlap in toe proportions and trackway parameters. Considering the small sample size (in terms of trackways per locality) their combination in the Birkheide–Gottlob and Hainesi–Willsi morphotypes has to be considered as a preliminary assessment. Thus, we have refrained from splitting "*Ichniotherium* with relatively short pedal digit V" into several ichnospecies—despite our interpretation that tracks assigned to the three morphotypes were produced by functionally distinct trackmakers. Our redefinition has also the benefit that the name *I. cottae* can be preserved for the Bromacker sample as the best known and largest individual sample of such trackways. However, studies that make use of the *I. cottae* data presented here or elsewhere should avoid their unreflecting inclusion as one homogeneous ichnospecies.

With the exception of the sparse record from the *I. cottae* type locality Gottlob quarry, this approach includes only true trackways, i.e., series of imprints that constitute at least one complete step cycle. However, our analysis also found moderate differences in the toe proportions suggesting that imprint morphology yields information useful for the subdivision of *I. cottae*. Given that other localities yield notable records of isolated imprints and imprint pairs (e.g., Czech Republic, *Fritsch, 1887*, *Pabst, 1908*; Morocco, *Lagnaoui et al., in press*; New Mexico, *Voigt & Lucas, 2015, in press*), imprint-morphology-based schemes could be tested through measurements on material not considered in the present approach and distributions for additional measures, such as angles between toes, total imprint width, length of the heel and depth in different parts of an imprint (which may be controlled by limb function, see *Romano, Citton & Nicosia, 2016*) could be gained. A feasible alternative to length-and-angle-based approaches would be a geometric morphometric analysis of imprint morphology.

## Phylogenetic trends in the producer group

Age and morphology of the Gottlob–Birkheide type are intermediate between the Hainesi–Willsi and Bromacker types of *I. cottae*. Thus, evolution of trackmaker locomotion might be inferred from changes in the trackmaker pattern from the earliest through the youngest morphotype, i.e., in a stratigraphic approach (Fig. 17). However, we consider a phylogenetic approach as better suited for a large geographically widespread set of individual ichnofossil records as it allows the inclusion of further types of overlapping age in the future (e.g., for the Maroon record). Furthermore, a phylogenetic hypothesis enables us to include *I. sphaerodactylum* and *I. praesidentis* as an outgroup to the assemblage of *I. cottae* morphotypes and to relate functional change inferred from tracks to diadectomorph phylogeny (Fig. 18).

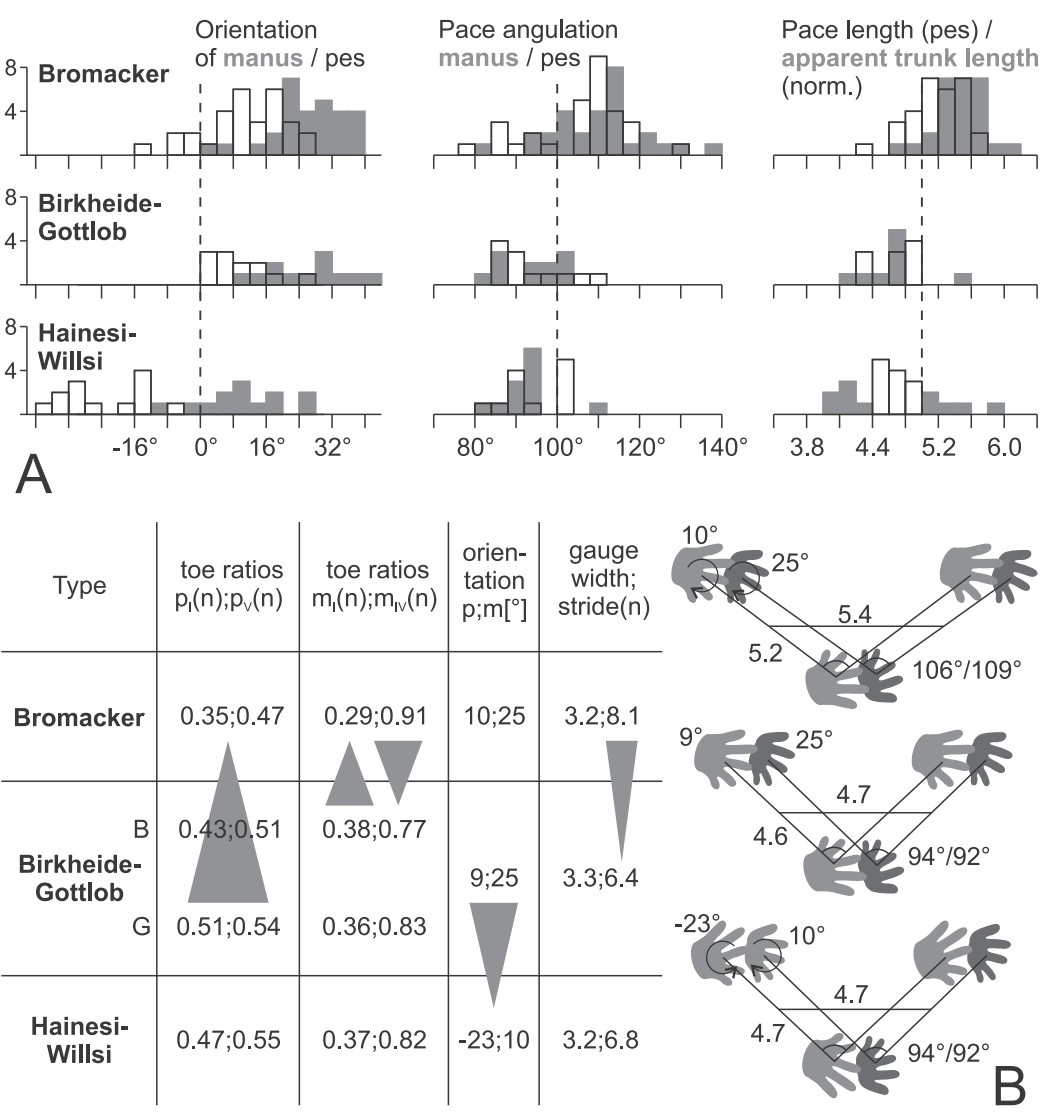

**Figure 17 Data for the three proposed morphotypes of *Ichniotherium cottae* in stratigraphic order may reflect changes in trackway pattern over time.** (A) Histograms depicting the distributions of six trackway parameters for each of the three types (number of counts = step cycles per bin). (B) Averages of imprint and trackway parameters. The sample from Gottlob quarry (G, the type locality of *Ichniotherium cottae*, Goldlauter Formation) lacks a sufficient number of complete step cycles, so only toe proportions of the Gottlob sample have been considered independently from those of the Birkheide record (B, Oberhof Formation) in the left part of the table.

Despite considerable differences between the Late Carboniferous *I. praesidentis* from the German Ruhr area and the Early Permian representatives of *Ichniotherium* from the Thuringian Forest, the presence of five manual digits, general similarity of the pedal imprint morphology and large body size are in good agreement with the assignment of *I. praesidentis* to diadectomorph producers (*Voigt & Ganzelewski, 2010*). Following the hypothesis that the *I. praesidentis* trackmaker either represents a member of diadectomorphs or another reptiliomorph group closely related to the Diadectomorpha–Amniota clade ("Cotylosauria" sensu *Laurin & Reisz, 1999*), the

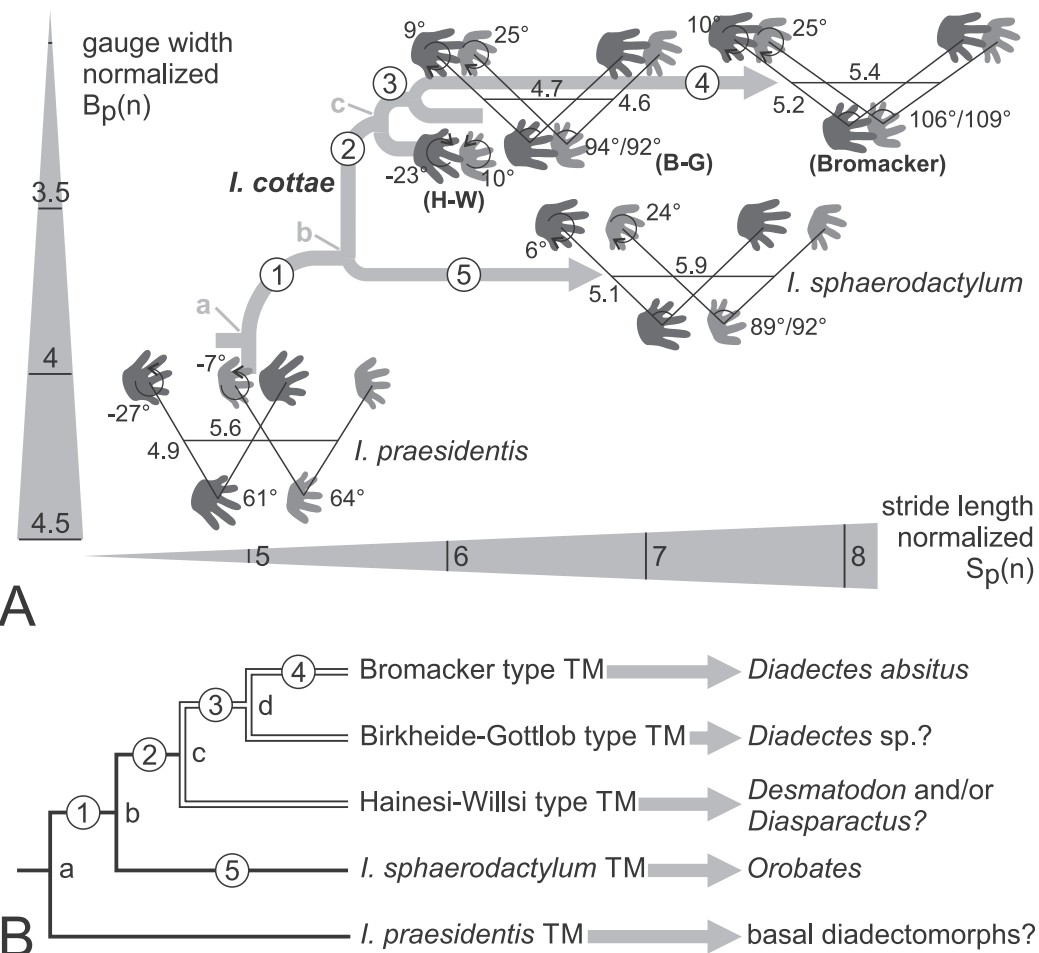

**Figure 18 Phylogenetic hypotheses for the trackmakers (TMs) of the *Ichniotherium* ichnospecies and morphotypes.** (A) Tree-like depiction of the evolutionary changes of *Ichniotherium* trackway patterns within the morphospace of body-size normalized gauge width vs. stride length. Numbers on trackway diagrams refer to pedal/manual imprint orientation angles, pedal/manual pace angulation, normalized pedal pace length and apparent trunk length. For the Hainesi–Willsi (H–W) type of *I. cottae* only imprint orientations differ notably from that of the Gottlob–Birkheide (G–B) type and are depicted here. Average values for *I. sphaerodactylum* and *I. praesidentis* are based on *Voigt (2007)* and *Voigt & Ganzelewski (2010)*, respectively. Evolutionary steps no. 1–2: decrease in normalized gauge width and distance from pedal to manual imprint of an imprint pair (in the direction of movement), increase in pace angulation and in normalized stride length, orientation of the manual imprints changes from outward (supination) to midline-parallel; no. 3: transition in pedal imprint orientation from outward to midline-parallel and in manual imprint orientation from midline-parallel to inward (pronation); no. 4: increase in pace angulation, in normalized pace length and stride length; no. 5: increase in pace angulation and in normalized stride length, orientation in pedal imprints changes from outward to midline-parallel, in manual imprints change from midline-parallel to inward orientation. (B) Phylogeny for all considered *Ichniotherium* TMs with nodes (a–c) and evolutionary steps 1–5 labeled as in (A). The arrows marks supposed correlations of ichnotaxa with fossil tetrapod orthotaxa. In accordance with the phylogenetic hypotheses of *Reisz (2007)* and *Kissel (2010)* node a might represent the last common ancestor of all diadectomorphs whereas node (b) represents the last common ancestor of *Orobates*, *Diadectes* and its allies. Node (c) corresponds to the *Desmatodon–Diasparactus–Diadectes* clade. Node (d) could represent the *Diadectes* lineage of *Reisz (2007)* and has no clear correspondence in *Kissel's (2010)* hypothesis.

occurrence of "*Ichniotherium willsi*," "*Baropus hainesi*" and "*Megabaropus hainesi*"
in the Late Carboniferous of England and Ohio—which are referred to the Hainesi–Willsi
type of *I. cottae* here—marks a notable transformation in the trackway pattern from a
presumably *Ichniotherium-praesidentis*-like last ancestral state (node (a) in Fig. 18): The
increase in stride length and pace angulation occurs in combination with a decrease in
trackway gauge width and apparent trunk length, reflecting an evolutionary change in
early diadectomorph locomotion (evolutionary steps 1 and 2 in Fig. 18).

We have proposed that *Ichniotherium* with relatively short pedal digit V
(ratio V/IV < 0.6)—which has been synonymized with *I. cottae* here—represents a
monophyletic producer group (corresponding to node (C) in Fig. 18) within Diadectidae
and includes Late Carboniferous and Permian taxa more closely related to *Diadectes* than
*Orobates* (see also Fig. 2). The hypothetical track type of the last common ancestor of
*Diadectes* and *Orobates* (corresponding to node (b) in Fig. 18) can be assumed to display
higher pace angulations, narrower gauges, higher stride lengths and more parallel-to-
midline orientated manual imprints than *I. praesidentis* (evolutionary step 1 in Fig. 18)
but it might have shared the plesiomorphic condition of outward-rotated pedal imprints
with *I. praesidentis* and the Hainesi–Willsi type of *I. cottae* whereas a relatively long pedal
digit V and high apparent trunk length might have been shared with *I. praesidentis* and
*I. sphaerodactylum*.

While average trackway gauge is comparatively narrow among all *I. cottae* tracks and
the manual and pedal imprints of an imprint pair are consistently set at a low distance
along a trackway (evolutionary step 2, possible synapomorphies for node (C) in Fig. 18),
several notable changes can be inferred when the three types of *I. cottae* are compared: The
Permian Gottlob–Birkheide and Bromacker types can be distinguished from the Late
Carboniferous Willsi–Hainesi type by their parallel-to-midline orientation of pedal
imprints and slight inward rotation of manual imprints (evolutionary step 3,
synapomorphy for node (d) in Fig. 18). Trackways of the best known last occurring type of
*I. cottae* from the Bromacker locality display larger pace angulations and body-sized-
normalized stride lengths than all other considered *Ichniotherium* samples (evolutionary
step 4 in Fig. 18). Apart from differences in the trackway pattern, slight differences in the
toe proportions occur (Figs. 12 and 14A and 14B).

Pinpointing these changes among *I. cottae* tracks to diadectid phylogeny is mostly
guesswork: If the assignment of the Bromacker *I. cottae* to *Diadectes absitus* according
to *Voigt, Berman & Henrici (2007)* is correct, the trackmaker of the Birkheide and
Gottlob trackway occurrences could either be a functionally different relative of *D. absitus*
(a representative of the *D. absitus* lineage) or else a relative of the North American
*Diadectes* species (*Diadectes* sensu *Kissel, 2010*) which was replaced by *D. absitus* in
the Thuringian Forest area before the deposition of the Tambach Formation
(stratigraphic level of the Bromacker site). Trackmakers of the Willsi–Hainesi type
might be found among contemporaneous Late Carboniferous diadectid species,
such as *Desmatodon hesperis* or *Diasparactus zenos* (*Reisz, 2007*; *Kissel, 2010*; see Figs. 2
and 18B).

Notable differences between the supposed lineages of *I. cottae* and *I. sphaerodactylum* trackmakers are the latter's high apparent trunk length, moderately wide gauge and relatively high distance between manual and pedal imprints of a couple (no overstepping). Notwithstanding these characteristics *I. sphaerodactylum* reaches body-size-normalized pace lengths and stride lengths that can be higher than in the Hainesi–Willsi and Gottlob–Birkheide types (see evolutionary step 5 in Fig. 16). Furthermore, *I. sphaerodactylum* shares with the Gottlob–Birkheide and Bromacker type tracks of *I. cottae* a more inward rotation of the pedal and manual imprints, a configuration that we consider as an independent parallel acquirement in both lineages of *Ichniotherium* trackmakers.

## Functional implications

Based on the comparison of average trackway patterns many aspects of the evolutionary change in body shape, posture and locomotion from an *Ichniotherium-praesidentis*-like last common ancestor of all *Ichniotherium* trackmakers towards the last common ancestor (and earliest occurring specimens) of *I. cottae* ("all *Ichniotherium* with relatively short pedal digit V") trackmakers can be deduced (evolutionary steps 1 and 2 in Fig. 18): The differences in gauge width, apparent trunk length and imprint orientations signify a decrease in the degree of sprawling, a proportional shortening of the trunk and more inward rotation (pronation) of the hands during ground contact. Higher strides (>20% increase in body-size-normalized values) and notably higher pace angulations (>25° increase) imply an increase in speed and general walking capability.

In the Early Permian Birkheide–Gottlob and Bromacker morphotypes of *I. cottae* the inward orientation of the manual imprints is more pronounced than in the Late Carboniferous Hainesi–Willsi type and accompanied by a considerable parallel to inward orientation of pedal imprints, suggesting a further change in hindlimb posture (evolutionary step 3 in Fig. 18) following the earlier decrease in sprawling. Another change—the concurrent increase in stride length (>10%), pace angulation (>10°), pace length and apparent trunk length towards the mid-Early Permian Bromacker type (evolutionary step 4 in Fig. 18)—probably represents a further speed increase. Unlike the differences in apparent trunk length found between the *Ichniotherium* ichnospecies this particular increase is arguably not indicative for an actual increase in the trackmaker's trunk proportions but rather due to the correlation of speed, stride and apparent trunk length in otherwise similar trackmakers (see also dependence of stride and glenoacetabular length according to *Leonardi (1987)*). A conspicuous difference between the Bromacker type and other *I. cottae* types does also occur in the imprint proportions: Relatively shorter pedal digits V and I and a shorter manual digit I in the Bromacker type might either be anatomically controlled, i.e., reflect actual variation in trackmaker toe proportions, or can be explained by reduced rotational movements of the autopodia on the ground or a changed center of rotation compared to earlier *I. cottae* types.

*Ichniotherium cottae* and *I. sphaerodactylum* from the Bromacker locality (see specimen UGKU 130 which features both types on the same slab) share higher normalized average

pace and stride lengths than earlier *I. cottae* types, indicating a convergent speed increase in the lineage of *I. sphaerodactylum* trackmakers (evolutionary step 5 in Fig. 18) after the divergence of the two ichnospecies (node (b)), which we pinpoint to the Late Carboniferous *Orobates–Diadectes* split. The trackmakers of Bromacker *I. sphaerodactylum* (i.e., *Orobates* and/or related diadectids) represent walkers with a somewhat longer trunk that was stabilized by a wider gauge and longer external toes than in the *I. cottae* trackmakers (see also *Voigt, Berman & Henrici, 2007*). The co-occurrence of two diadectid trackmakers within derived locomotion capabilities but differences in body shape and posture may be explained by different foraging strategies or occupation of different (but overlapping) sub-environments (*Marchetti, Voigt & Santi, in press*).

In sum, we observe a clear pattern of evolutionary change in terrestrial gait within (a) diadectid trackmakers of *I. cottae* as an ichnotaxon composed of different forms that share a relatively short pedal digit V and (b) *Ichniotherium* trackmakers as a whole. Following an early phase of evolutionary change in posture and locomotion with a reduction in the degree of sprawling, shortening of the trunk, slight pronation of the hands during ground contact and somewhat higher walking speeds, the later transformation of the trackway pattern towards a more pronounced inward orientation of the manual and pedal imprints indicates a further change in posture which arises in combination with a further increase in walking speed.

Diadectid evolution is thought to have been shaped by adaptation to a herbivorous lifestyle which is visible in the phylogenetic transformation of the skull towards a higher ability of processing plant material, but also in the gaining of body sizes that are not matched by more basal carnivorous terrestrial tetrapods (*Sues & Reisz, 1998*; *Reisz & Sues, 2000*; *Kissel, 2010*; *Reisz & Fröbisch, 2014*). In conflict with a late or continuous increase in *Ichniotherium* trackmaker body size, the earliest occurrences from the Late Carboniferous and Carboniferous/Permian boundary include the largest individuals, i.e., *I. praesidentis* from Bochum (pes length up to 200 mm; *Voigt & Ganzelewski, 2010*; *Schöllmann et al., 2015*), two tracks from Haine's Farm (147–186 mm) and the Marietta specimen (156–183 mm, see Supplemental Information S3). In this regard, a herbivory-related body size increase in Late Carboniferous diadectomorphs might have been an evolutionary step that initiated later changes in locomotion or released constraints on terrestrial mobility, but it does not explain the further speed increase and postural change towards the medium-sized Bromacker diadectids (pes length 67–88 mm in the Bromacker type of *I. cottae*, 82–136 mm in *I. sphaerodactylum*; *Voigt, Berman & Henrici, 2007*). Considering this study's focus on angle measurements, (dimensionless) length ratios and a certain group of *Ichniotherium* tracks, the question of body size evolution in trackmakers surrounding the origin of amniotes cannot be exhaustively dealt with here (and shall be discussed elsewhere).

It should be noted that phylogenetic and functional implications discussed above share the problems of our ichnotaxonomic assessments: They are based on a relatively low number of specimens per locality and on a limited number of localities (those with actual trackways of *Ichniotherium* and not only incomplete step cycles). Most of the trackways discussed here come from the Bromacker and Birkheide quarries close to the town of

Tambach-Dietharz (Thuringian Forest). Problems with the classification of certain *Ichniotherium* samples from localities in the United States and Poland suggest that a more complex picture will arise in the future with an increasing non-European record and higher overall sample size. Some of our conclusions regarding the evolution of function depend on the hypothesis that *Ichniotherium* with relatively short pedal digit V actually corresponds to a monophyletic group of diadectid producers which share a short fifth pedal toe as a synapomorphy. Osteological data supporting alternative scenarios of pedal toe reduction in diadectids would weaken our hypothesis of evolutionary advance within the group of *I. cottae* trackmakers.

In accordance with earlier phylogenetic approaches to tetrapod tracks (*Carrano & Wilson, 2001*; *Wilson, 2005*) our phylogenetic interpretation of *Ichniotherium* trackways relies on synapomorphy-based and stratigraphy-based correlation assumptions in addition to direct track-trackmaker correlation (Bromacker site). Following the idea that trackway data yield information on tetrapod locomotion that cannot be deduced from skeletons, mapping trackway data on a phylogenetic tree of trackmakers opens up a way to infer evolution of locomotion styles and related functional traits—if the trackmaker identity can be assigned with confidence. Arguably this way of reasoning about Paleozoic tetrapod tracks, especially the reconstruction of ancestral states for discrete or continuous trackway characters (see e.g., *Cunningham, Omland & Oakley, 1998*), can help to solve questions of locomotion evolution surrounding the origin of amniotes in the future.

## CONCLUSION

Measurements of 10 toe lengths and six independent trackway parameters have been carried out for a sample of 25 *Ichniotherium* trackways (69 step cycles) from nine localities. Based on locality-wise quantitative comparisons of these trackways, three morphotypes of "*Ichniotherium* with relatively short pedal digit V"—the Birkheide–Gottlob, Bromacker and Hainesi–Willsi type—have been distinguished and related to certain functionally distinct diadectid trackmakers more closely related to *Diadectes* than *Orobates*. Given the small overall sample size and remaining uncertainties in the distinction of the three types, we suggest the use of the ichnospecies *I. cottae* (*Pohlig, 1892*) for all "*Ichniotherium* with relatively short pedal digit V." Including the three types of *I. cottae* in a phylogenetic framework together with *I. sphaerodactylum* and *I. praesidentis*, a trend of evolutionary advance in locomotion from the last common ancestor of all *Ichniotherium* trackmakers to the last common ancestor of all *I. cottae* producers and from the latter to the trackmakers of the mid-Early Permian Bromacker type can be deduced. Among others, evolutionary transformation in trackmaker locomotion is reflected by the occurrence of more parallel to inward manual and pedal imprint orientations, more obtuse pace angulations, narrower gauges, higher body-size-normalized pace lengths and higher body-size-normalized stride lengths. Since they have mainly been inferred based on European trackways records, these changes might either represent a local signal or a more general pattern of diadectomorph evolution.

## INSTITUTIONAL ABBREVIATIONS

| | |
|---|---|
| **BU** | Lapworth Museum of Geology, University of Birmingham, Great Britain |
| **CMNH** | Cincinnati Museum of Natural History, Cincinnati, USA |
| **DMNS** | Denver Museum of Nature and Science, Denver, USA |
| **HF** | Institut für Geologische Wissenschaften und Geiseltalmuseum, Martin-Luther-Universität Halle-Wittenberg, Germany |
| **KGM** | Kletno Geological Museum, Poland |
| **MB** | Museum für Naturkunde, Berlin, Germany |
| **MC** | Marietta College, Ohio, USA |
| **MNG** | Museum der Natur Gotha, Germany |
| **MSEO** | Museum Schloss Ehrenstein, Ohrdruf, Germany |
| **NHMS** | Naturhistorisches Museum Schloss Bertholdsburg, Schleusingen, Germany |
| **OSU** | Orton Geological Museum, Ohio State University, Columbus, USA |
| **PMJ** | Phyletisches Museum Jena, Germany |
| **SSB** | Sammlung Stober, private collection, Berlin, Germany |
| **UGKU** | POLLICHIA Geoscience Collections, Urweltmuseum Geoskop, Burg Lichtenberg, Thallichtenberg, Germany |

## ACKNOWLEDGEMENTS

We are thankful to Gottfried Böhme, Peter Cramer, Dale Gnidovec, Hartmut Haubold, Andrea Heinke, Wolf-Dieter Heinrich, Jason Hilton, Dietrich von Knorre, Thomas Martens, Arnold Niziołek, Gerd Riedel, Dieter Schweiss, Bryan Small, Hans-Detlef Stober, Glenn Storrs, Ralf Werneburg and Frederick Voner for access to the specimens included in this analysis. Kirstin S. Brink, Marco Romano, Peter Falkingham and Jens Lallensack are gratefully acknowledged for their critical comments on an earlier version of this manuscript.

### Funding

The authors received no funding for this work.

### Competing Interests

Michael Buchwitz is an employee of the city administration of Magdeburg, the state capital of Saxony-Anhalt. Sebastian Voigt is an employee of the Pfalzmuseum für Naturkunde/Pollichia Museum.

### Author Contributions

- Michael Buchwitz conceived and designed the experiments, analyzed the data, contributed reagents/materials/analysis tools, wrote the paper, prepared figures and/or tables, reviewed drafts of the paper.
- Sebastian Voigt conceived and designed the experiments, performed the experiments, contributed reagents/materials/analysis tools, wrote the paper, prepared figures and/or tables, reviewed drafts of the paper.

## Data Availability

The raw data has been supplied as Supplemental Dataset Files.

## Supplemental Information

Supplemental information for this article can be found online at http://dx.doi.org/10.7717/peerj.4346#supplemental-information.

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
