# Peer review of "On the morphological variability of Ichniotherium tracks and evolution of locomotion in the sistergroup of amniotes"

_PeerJ, doi:10.7717/peerj.4346_

## Round 0.1 · original submission · Major Revisions

Thank you for submitting this interesting paper on fossil trackways and their nuances. I agree with the reviewers that this would be a nice contribution to the literature. There are numerous constructive critiques from the two reviewers that will need to be addressed (with a point-by-point rebuttal), and at least brief re-review. We look forward to the revised MS.

·

Basic reporting

Dear Editor
I reviewed the manuscript entitled “On the morphological variability of Ichniotherium tracks and locomotion upgrading in the sistergroup of amniotes” by Michael Buchwitz and Sebastian Voigt. The manuscript represents an interesting quantitative approach to a very well-known ichnotaxon from the Late Carboniferous to Early Permian ichnological record. However, some weaknesses in the methods used may have a greater impact on the interpretations and conclusion of the work. Below some indications and discussions are reported that must be carefully considered before the manuscript can be considered for publication. I'm not a native speaker, but English seems correct and very fluent.
Below, some methodological considerations and other more theoretical ones are reported, which I hope can improve the manuscript and the conclusions obtained and presented.

Experimental design

-Concerning the Principal Component Analysis, from the dataset results, as usual, the presence of several missing entries. It would be useful to specify how the missing values were handled in the analysis. In PAST the default modality to replace missing entries is the ‘mean value imputation’: however it is largely preferable to use the ‘iterative imputation’ algorithm (Hammer 2013). For the iterative imputation the missing values are replaced by their column average in a first stage; a PCA analysis is then run and used to compute regression values for the missing data, and this procedure is simply iterated to reach convergence (see Hammer 2013). If this algorithm was not used, it is strongly recommended to try to redo some analyses to see if significant different results are obtained.

-In the same way, if the raw data were not log-transformed before the analysis, misleading results can be obtained. Transformation into logarithm is strongly recommended to fit linear models and for the correspondence of the log-transform to an isometric null hypothesis (see Chinnery, 2004; Cheng et al., 2009; Romano and Citton, 2015, 2016; Romano, 2017). Again it is recommended then to perform the analysis only after having log-transformed the raw data.

-It would be useful to show the loadings of the single principal components, and describe briefly to what the greatest variance in each component is linked. If, for example, the length of digit IV contributes consistently to the variance (for convention loadings values greater than 0.3 and lower than -0,3) the use of this length to normalize the others results in a both methodological and conceptual error.
It might also be useful to show the scatter plot of components 2 vs 3 (and all other components with significant variance), which very likely are more related to morphometry of the footprints and trackways parameters rather to absolute dimensions. Usually is fine to show only the first two principal components if they describe more than 95% of the detected variance. However, in the submitted work the first two components describe on average only 80% of the variance; thus is there an additional 20% of variance in other components which might be useful to show, and perhaps lead to different discussions and conclusions. In the case of PCA performed on the six trackway parameters, the two first principal components account for only about 63% of the variance. It is therefore necessary to show how the remaining 40% is distributed and if the scatter plots considering the other components provide different result.
An analysis of PCAs using digit lengths in manus and pes of Ichniotherium has already been conducted recently, and described in detail in Romano and Citton (2015). It might be worth mentioning briefly this work and comparing the results with the new results.

-One element that can bring to weak results is the choice of the homologue points for digit measurements. In the manuscript, the free digits length is used, a fairly weak character that can change consistently even within the same trackway (pers. obs.), due to factors both directly related to the mode of locomotion and interaction between substrate and autopods and of a purely taphonomic nature (substrate conditions and relative footprint preservational type). It is recommended otherwise to use the phalangeal length of the digit (see, Leonardi, 1987; Romano and Citton, 2015, Fig. 1). In fact, the most useful method to obtain anatomical information about the trackmaker is the measure of the digit phalangeal portion length, concerning ‘…the measure of the segment that joins the distal extremity of the digit with the corresponding mid-point of the metapodial-phalangeal pad’ (Leonardi, 1987, p. 47), which is more closely related to the real anatomical length of the digit. If the metatarsal-phalangeal pad is not visible or preserved, it is advisable not to consider such footprints in the analysis. In fact, is better to have fewer footprints but definitely linked to a real osteological pattern and structure, rather than many tracks where digit lengths are extremely variant, and not depending on the trackmaker's anatomy. It is likely that, by decreasing the number of tracks to the best and anatomically significant footprints, the distance between the three recognized morphotypes in I. cottae reduces or totally disappears.

-Another possible problem is that the PCAs were performed not on linear measures, but on ratios between measures (in particular the various lengths normalized for digit IV). First of all, it is strongly advised to not use ratios at all in principal component analyzes (see Hammer and Harper, 2006); ratios are best indicated, differently, for classical cluster analyses. By dividing all the measures by the length of digit IV (or by other lengths), the necessary independence of the different variables is lost (a fundamental requirement for classical PCA), with the possible obtainment of misleading trends or clusters (especially if the digit chosen as reference is variant or peculiar). In fact, normalizing the same measure can lead to more uniform and separate groups, but nevertheless artificial. The same methodological problem applies to track parameters included as a ratio, and normalized for the length of digit IV (or other lengths). According to Hammer and Harper (2006) “Compositional data (relative proportions) must be analyzed with a special form of PCA (Aitchison 1986, Reyment & Savazzi 1999), because we need to correct for spurious correlations between the variables”. The biggest problem is indeed the “spurious correlations”, with detection of artificial clusters due to normalization, and to the non-independence of variables.

-Since in the work a sort of evolution in posture and in trackways parameters is described, with Ichniotherium preaesidentis as outgroup, followed by I. sphaerodactylum, I am surprised that these taxa have not been included in the PCA analyses and in the other graphs of figures 10 and 12. Such an analysis (by using linear measures, as already indicated, and not ratios) it could show how the three recognized morphotypes for I. cottae result closer to each other than to I. praesidentis and I. spaherodactylum. If that is not the case, it means that the length of the fifth digit is not a solid and sufficient character for the division of ichnotaxa. Since the authors speak of phylogeny, it is good to remember how the use a single character is typical of the nineteenth-century pre-cladistic typological philosophy. Starting with Hennig, if one want to talk about phylogeny, is necessary to consider the congruence tests of a complex set of characters (for example under the principle of parsimony or other principles), and not of individual characters (see Romano and Nicosia, 2015).

Validity of the findings

-An important element to be discussed is the ichnotaxonomic consequences of the study. According to the authors, within the specimens referred to Ichniotherium cotte (on the basis of a relatively short digit V) three distinct morphotypes are recognizable, in particular the 'Birkheide-Gottlob type', the 'Bromacker type' and Hainesi-Willsi type'. Again according to the authors, each of the morphotypes shows a functionality and locomotion substantially different, therefore attributable to three different trackmakers (or evolutionary degrees in the conclusions). A first crucial point on the philosophical level is the problem to include three different recognized biological trackmakers under the same ichnospecies; especially if sufficient characters are available to even distinguish evolutionary degrees in locomotion and functionality from trackways parameters. The question could be as follows: if such a high number of track parameters to distinguish three morphotypes and different functionalities are available, why not set up three separate ichnotaxa? From what has been reported and discussed in the paper, seems that the fundamental feature (the “synaphomorphy”) characterizing Ichnotherium cottae is simply a relatively short digit V. However, the length of a single digit, even at the level of genetic plasticity, is an extremely variant element in evolution, with loss and addition of phalanxes in repeated and paraphyletic ways. It seems that this character on which all the I. cottae “box” is founded, results weaker than a large number of trackways parameters, best referable to different trackmakers. If the ichnotaxonomy, in a new illuminated vision, must be based on natural biological taxa (i.e. zoological trackmaker and their unique combination of synapomorphyc, plesiomorphyc and autapomorphyc characters; see Olsen, 1995; Carrano and Wilson, 2001; Romano et al., 2016), it is difficult to imagine that different degrees of evolution in locomotion an functionality (referable to different trackmakers as found in the work) are hierarchically subordinate to the number of phalanxes on a single digit, within the framework of the hierarchic enkaptic system founded by Willi Hennig.
Secondly, one may wonder if having such conservative autopod structures in all morphotypes, we are dealing just with a same kind of biological trackmakers in different ontogenetic conditions, and under different types of locomotion. Simply changing the speed, trackways parameters (as experimentally observed) change accordingly, with trackway light that narrows, stride that increases, axes of autopods more parallel to the direction of advancement and so on. If the various track parameters change together (strong covariance) from a morphotype to another (what seems to happen), it would be more parsimonious to imagine a change in the locomotion and speed of the same trackmaker type. After having seen the tremendous difference between a normal sprawling crocodile and the same individual under a galloping locomotion, I think that the simple trackway parameters must be taken with extreme caution in obtaining greater conclusions in the macro-evolutionary field.
Another element that could be taken into consideration is the differential depth of the different portions of the footprints, as an evidence to infer the trackmaker biomechanics. In fact, as shown in some analyses performed also on Ichnotherium material (see Romano et al. 2016), such a study can shed light on the various functional axes active from time to time during stroke progression, and highlight synapomorphyc characters of a particular vertebrate clade, mirrored in the preserved tracks. So, it would be interesting to know whether in the three recognized morphotypes within Ichniotherium cottae, the differential depth of various footprint portions is uniform during the three main phases (‘touch-down’, ‘weight-bearing’ and ‘kick-off’ phases), or if substantial changes in the distribution of pressures are observed (and therefore in biomechanics and functionality that can be inferred).

The authors are free to contact me for any points raised on this review
kind regards
Marco Romano
Berlin
11/10/2017


Literature mentioned
Aitchison, J. 1986. The Statistical Analysis of Compositional Data. Chapman & Hall, New York.
Carrano, M.T., Wilson, J.A., 2001. Taxon distributions and the tetrapod track record. Paleobiology. 27, 564‒582.
Cheng, Y.N., Holmes, R., Wu, X.C., Alfonso, N., 2009. Sexual dimorphism and life history of Keichosaurus hui (Reptilia: Sauropterygia). J. Vertebrate Paleontology
29, 401e408.
Chinnery, B., 2004. Morphometric analysis of evolutionary trends in the ceratopsian postcranial skeleton. J. Vertebrate Paleontology 24, 591e609.
Hammer Ø. 2013. PAST Paleontological Statistics Version 3.0: Reference Manual. University of Oslo.
Hammer, Ø.& Harper, D.A. T. 2006. Paleontological Data Analysis. Oxford: Blackwell Publishing Ltd, 351 pp.
Liu, J., & Bever, G. S. 2015. The last diadectomorph sheds light on Late Palaeozoic tetrapod biogeography. Biology letters, 11(5), 20150100.
Olsen, P.E. 1995. A new approach for recognizing track makers. Geol. Soc. Am. Abs. Prog., 27, 86.
Reyment, R.A. & E. Savazzi. 1999. Aspects of Multivariate Statistical Analysis in Geology. Elsevier, Amsterdam.
Romano, M., 2017. Long bone scaling of caseid synapsids: a combined morphometric and cladistic approach. Lethaia. http://dx.doi.org/10.1111/let.12207.
Romano, M., & Citton, P. 2015. Reliability of digit length impression as a character of tetrapod ichnotaxobase: considerations from the Carboniferous–Permian ichnogenus Ichniotherium. Geological Journal, 50(6), 827-838.
Romano, M., & Nicosia, U. 2015. Cladistic analysis of Caseidae (Caseasauria, Synapsida): using the gap‐weighting method to include taxa based on incomplete specimens. Palaeontology, 58(6), 1109-1130.
Romano, M., Citton, P., & Nicosia, U. 2016. Corroborating trackmaker identification through footprint functional analysis: the case study of Ichniotherium and Dimetropus. Lethaia, 49(1), 102-116.

Additional comments

-In Figure 1C the last occurrence of diadectids is made to correspond to the Kungurian. However, the new diadectid Alveusdectes fenestralis has been recently described from the Upper Permian of China (Liu and Bever, 2015). It may be useful to update the scheme using the new cladogram provided by Liu and Bever (2015, fig.2, p. 4), and extending the ‘Diadects and allies’ branch up to the Wuchiapingian.

-In the text is highlighted “a surprising difference between the Bromacker sample and older trackways from the Birkheide and Gottlob localities”. However, two of the three specimens from Gottlob in Figure 10 C fall completely into the morphospace of Bromacker. Moreover, an overlap is also present with the Birkheide specimens in Figure 10 D and again with the ones from Gottlob in Figure 10 E.

·

Basic reporting

The English language is generally good, but there are a few instances that should be fixed. Some examples where the language could be improved include:
-Title: What is ‘locomotion upgrading’ ? This is an awkward statement that is used repeatedly in the manuscript. Maybe change it to be an increase in efficiency? A suggestion for the title is: “on the morphological variability of Ichniotherium tracks and the evolution of locomotion in the sistergroup of amniotes”
-“number of couples” appears throughout text and in supplementary data… does this mean imprint pairs?
-Line 102: group of all yet undocumented Ichniotherium…
-Line 163: orientations towards walking direction… change to orientations in direction of movement
-Line 249: PC 1 and 2, not pr 1 and 2
-Line 255: Its more inward orientation… Change to The more inward
-Line 264: …distinction of THE Maroon sample…
-Lines 278-282 unclear phrasing. They are not distinct from each other? Or the rest of the sample?
-Line 290: …That does never fall below… not fall below
-Line 295: …causes much of overlap with the Birkheide… The overlap
-Line 297: ‘curve walk’ is awkward
-Line 304: direction of movement, not walk
-Line 333: ‘morphologically close’ change to morphologically similar.
-Lines 427-430: This sentence is very long and awkward.
-Line 455: hypothetical
-Line 542-3: …of the skull towards higher ability of processing plant material…
-Line 562-564: awkward sentence
-Throughout the manuscript: Ichinotherium is abbreviated to I. inconsistently.

Line 544: add this more recent reference: Reisz, R.R., and Fröbisch, J. 2014. The Oldest Caseid Synapsid from the Late Pennsylvanian of Kansas, and the Evolution of Herbivory in Terrestrial Vertebrates. PLoS ONE 9(4): e94518.

Experimental design

The rationale for the study is not clearly stated. The abstract says: 'Here we document variation of digit proportions and trackway parameters….' Why? Lines 104-123 state the steps that were taken to perform the study, but lines 104-105 should state the question/problem being addressed. Is to document if all tracks attributed to I. cottae are in fact I. cottae? Is it to see if all tracks of I. cottae can be attributed to the same trackmaker? Are you only focusing on I. cottae because I. sphaerodactyulm and I. praesidentis are less variable, or have a smaller sample size? Why would you expect I. cottae to belong to separate trackmakers? I think the paper would be much stronger with a directed rationale for the study and a justification for the focus on I. cottae.

I have one concern with the statistical analyses. The inclusion of angle measurements and linear measurements together in the PC analyses might be problematic, as the angles are circular and the other measurements are linear data. I am unsure if both types of data (with different units) can be combined in a PCA. It is possible that transforming the angle measurements, or a separation of angle and linear measurements for the PC analyses will avoid any potential problems. There are many books that discuss combining linear and circular data in descriptive statistical analyses, for example:
Batschelet, E., 1981. Circular statistics in biology. ACADEMIC PRESS, 111 FIFTH AVE., NEW YORK, NY 10003, 1981, 388.
Berens, P., 2009. CircStat: a MATLAB toolbox for circular statistics. J Stat Softw, 31(10), pp.1-21
Jammalamadaka, S.R. and Sengupta, A., 2001. Topics in circular statistics (Vol. 5). World Scientific.
Mardia, K.V., 2014. Statistics of directional data. Academic press.

Validity of the findings

The findings are valid. Detail should be added to the figure caption for Fig. 16 explaining what is occuring at each 'evolutionary step' (1–5). It is described in-text but would be much clearer if the changes occuring at each step could be described in the figure caption. Is the phylogeny based on a previously published phylogeny for diadectids? Do nodes a-d correspond to any specifically named groups?

The findings should relate back to the question/problem being addressed (which is not clear in this paper).

Additional comments

Overall, the paper is very interesting. I like that the trackway record can be used to infer the evolution of locomotion in diadectids. The figures are nice and clear. With a few minor changes, the manuscript will be ready for publication.

---

## Round 0.2 · Major Revisions

Thank you for submitting the revised MS, which has improved. One reviewer only wants small final changes. The other still is not convinced, especially by the PCA methods and conclusions drawn from them (I agree with the reviewer's implication that using PCA to "illustrate differences" is still, in essence, hypothesis testing and formulating conclusions, even if they are qualified as speculative/exploratory (other readers might fairly easily take these speculations to be actual conclusions; it is a risk for the scientific literature). The reviewer seems to make fair points here, and it does not seem valid to me to address their points mainly in the Rebuttal than in the actual MS. I urge the authors to try to revise the MS itself to more thoroughly convince the reviewer and strengthen the MS, via more compromise. I recognize however that there may simply be some differences of opinion in the end but via diplomacy between the authors and reviewer I am hopeful that the MS can reach a better endpoint and then we could accept it. Apologies for any frustration this may cause.

·

Basic reporting

Please see attached PDF

Experimental design

Please see attached PDF

Validity of the findings

Please see attached PDF

Additional comments

Please see attached PDF

·

Basic reporting

Good

Experimental design

Good

Validity of the findings

Good

Additional comments

I think that all reviewer comments have been addressed appropriately and the paper is improved. My only suggestion is to include the rationale for the study in the abstract, and see attached file for other small corrections.

---

## Round 0.3 · Major Revisions

Apologies for the delay in reviews and decision. As there were lingering strong disagreements on the science between 1 reviewer and the authors, I had to find 1 additional reviewer, who broadly is convinced by the MS but has some final queries.

However, as that reviewer could not comment on the statistics, I had to consult another Associate Editor with expertise in PCA. I append their comments here. Please revise the MS accordingly and respond to all comments from reviewers and the Associate Editor. I will then render a final decision, but will attempt to minimize further delays-- as long as the authors, in good faith, show a good effort to address all points.

Appended editorial comments:
1) The methods need to more explicitly say what variables they used in the PCA and how many combinations of variables they included in the PCA(s). I cannot really figure this out by what is presented. Did they do 1 PCA on the Toe ratios and a 2nd PCA on the trackway measures (angles, pace etc)? Was there a 3rd PCA?

2) The reasons to log values: (a) if the measures are very different in scale. For instance, 0.5 is very different from 100. The 0.5 will load less as it is a smaller number, even if they are measuring similar things. (b) if the data are not normally distributed. (c) if there is a variance problem. (d) if the data are not linear. What reason does the Review give for logging the data?

3) But, you don't always have to play by these 'rules'.

2) I think the Reviewer is wrong on the log issue in point (1) by the Authors - if the PCA refers to the Toe Ratio data (which I presume is normalized digit lengths). By taking the proportion of the digits the authors have removed size and scale before running their PCA. This is accurate. Logging the raw values does not make sense if the proportions are the only thing being used in the PCA. The Reviewer will get size as the main source of variation on PC1 by just logging the raw length data as size has not been removed - only scale has been removed; the following PCs will also be different.

3) Logging raw values and running PCA and then neglecting PC1 because it is size is also wrong. By removing PC1 in this way you not only remove isometric shape, you remove allometric shape. By taking the ratio and then running the analysis, isometric shape was removed before the analysis but allometric shape was retained.

4) The Author's attempt at showing how logging the data were incorrect is actually wrong. If the data are to be logged it is the Ratio that should be logged not the raw data before taking the ratio. If the Ratio is logged it will be the same as the non-logged Ratio - just linear etc. This could be a possibility if there is a problem with the normality of the ratios. The Authors should do this to see if it changes the results. If the raw data are logged, a ratio should not be calculated as it has modified the data!

5) Another way to do the analysis would be to remove size and log normalize before the PCA. PAST has different options for this, but the ratio should be fine I think - unless the data are really non - normal.

6) If the Authors use different types of data in a combined PCA (e.g. angles, linear, ratio) I think they need to use the correlation matrix and not the coefficients matrix. I can't tell which one they used. It is fine to combine different types of data - the Reviewer is wrong. You just have to be careful how it is handled. A PCO could also be used (but not using the Euclidean matrix as that is the same as a PCA with coefficient matrix - if I remember correctly).

7) Using pedal pIV for the toe ratio also seems fine. It has the lowest variance which is good. However, by doing this the Authors cannot examine changes in pIV - as all specimens have the same variance. This means that if a specimen has a weirdly long or short pIV it would not be accounted for appropriately. Another way to do a ratio would be to divide by the geometric mean - but this would require all digits. I am not sure what proportion of the data is missing; such a tactic might not be possible. It might be that pIV is the best they can do with what they have.

8) I am not sure how useful pIV is for normalizing place length and trunk length. It all depends on if it is a good measure of size. Are animals with proportionally large pIV bigger in total size too?

8) In addition to the PCAs, I also notice that the Authors also present regular bi-plots of two variables against each other. The normalized linear measures (e.g. toe ratios or pace length) could be logged in this instance before running the analysis. I am not sure it will greatly affect the result - but it might be a way to deal with the Reviewer's obsession with logs!

·

Basic reporting

The text is professional, albeit extremely dense and likely impenetrable for many readers. The same can be said of some of the figures (esp. 1, 13, 15), which are crowded to the point of being difficult to read. While they could be made more accessible, to do so would require a significant re-write that I do not feel is necessary.

Experimental design

My main worry is that so many of the tracks figured have a very ambiguous morphology between digits and the rest of the foot, which makes reliable measures of digit length somewhat difficult. The authors do address this in the text (ln 203), but still many figured tracks do not show connected digits and 'palm/sole'. Perhaps my concerns could be addressed with a figure showing the measurements as taken on some of the more disconnected tracks.

Validity of the findings

The authors make a case for locomotor evolution observed in these tracks. Their conclusions are slightly cautious: "Since they have mainly been inferred based on European 625 trackways records, these changes might either represent a local signal...". I would like to see a little caution in the abstract too. While their conclusions are supported by their data, I think the vagaries of sample size and track morphology warrant more tentativeness.

Additional comments

Whilst I think it would be unreasonable to suggest a re-write to make the manuscript more accessible, there are places where sentences can be tidied up a little. For instance, lines 112-117 are a single sentence paragraph.

Ln 141, 230; It's not clear what the authors mean by 'true trackway'. Given that current ichnological terminology (Marty et al 2016) defines a 'true track' as the foot-sediment interface, or a direct track, it seems that this is not what the authors are referring to with 'true trackways'. If they are, it means all other tracks they measured are transmitted tracks or overprints, in which case measures of digit length loose meaning. If 'True trackway' refers to a specific sequence of footfalls, then the authors may wish to use a different term, e.g. 'complete step cycle'

ln 601: after 'functional traits' the authors should add "providing trackmaker ID can be made with confidence"

---

## Round 0.4 · accepted · Accept

The authors have done an excellent job handling reviews, with careful attention to detail. The Rebuttal is very well done with clear justifications. Hence I see no reason why there should be further review. If disagreements remain with any reviewers then they can be sorted out in post-peer review/literature. This seems like a very conscientious revised MS and so I am glad to accept it. Congratulations!